# MG-TSD: Multi-Granularity Time Series Diffusion Models with Guided Learning Process

**Xinyao Fan**[1*], **Yueying Wu**[2*], **Chang Xu**[4†], **Yuhao Huang**[3], **Weiqing Liu**[4], **Jiang Bian**[4]

University of British Columbia[1], Peking University[2], Nanjing University[3], Microsoft Research[4]

`xinyao.fan@stat.ubc.ca, wuyueying@stu.pku.edu.cn,`
`huangyh@smail.nju.edu.cn,`
`{chanx, weiqing.liu, jiang.bian}@microsoft.com`

## Abstract

Recently, diffusion probabilistic models have attracted attention in generative time series forecasting due to their remarkable capacity to generate high-fidelity samples. However, the effective utilization of their strong modeling ability in the probabilistic time series forecasting task remains an open question, partially due to the challenge of instability arising from their stochastic nature. To address this challenge, we introduce a novel **M**ulti-**G**ranularity **T**ime **S**eries **D**iffusion (**MG-TSD**) model, which achieves state-of-the-art predictive performance by leveraging the inherent granularity levels within the data as given targets at intermediate diffusion steps to guide the learning process of diffusion models. The way to construct the targets is motivated by the observation that the forward process of the diffusion model, which sequentially corrupts the data distribution to a standard normal distribution, intuitively aligns with the process of smoothing fine-grained data into a coarse-grained representation, both of which result in a gradual loss of fine distribution features. In the study, we derive a novel multi-granularity guidance diffusion loss function and propose a concise implementation method to effectively utilize coarse-grained data across various granularity levels. More importantly, our approach does not rely on additional external data, making it versatile and applicable across various domains. Extensive experiments conducted on real-world datasets demonstrate that our **MG-TSD** model outperforms existing time series prediction methods. Our code is available at `https://github.com/Hundredl/MG-TSD`.

## 1 Introduction

Time series prediction is a critical task with applications in various domains such as finance forecasting (Hou et al., 2021; Chen et al., 2018), energy planning (Koprinska et al., 2018; Wu et al., 2021), climate modeling (Wu et al., 2023; 2021), and biological sciences (Luo et al., 2020; Rajpurkar et al., 2022). Considering that time series forecasting problems can be effectively addressed as a conditional generation task, many works leverage generative models for predictive purposes. For instance, Salinas et al. (2019) utilizes a low-rank plus diagonal covariance Gaussian copula; Rasul et al. (2021) models the predictive distribution using normalizing flows. Recent advancements in diffusion probabilistic models (Ho et al., 2020) have sparked interest in utilizing them into probabilistic time series prediction. For example, Rasul et al. (2020) auto-regressively generates data through iterative denoising diffusion models. Tashiro et al. (2021) uses a conditional score-based diffusion model explicitly trained for probabilistic time series imputation and prediction. These methods relying on diffusion models have exhibited remarkable predictive capabilities. However, there is still considerable scope for improvement. One challenge that diffusion models face in time series forecasting tasks is the instability due to their stochastic nature when compared to deterministic models like RNNs and variants like LSTMs (Hochreiter & Schmidhuber, 1997; Lai et al., 2018), GRUs (Ballakur & Arya, 2020; Yamak et al., 2019), and Transformers that rely on self-attention mechanisms (Vaswani et al., 2017; Zhou et al., 2021; 2022; Wu et al., 2021). More specifically, the

---

*These authors contributed equally to this work.
† Corresponding to chanx@microsoft.com

diffusion models yield diverse samples from the conditional distributions, including possible low-fidelity samples from the low-density regions within the data manifold (Sehwag et al., 2022). In the context of time series forecasting, where fixed observations exclusively serve as objectives, such variability would result in forecasting instability and inferior prediction performance.

To stabilize the output of a diffusion model in time series prediction, one straightforward method is to constrain the intermediate states during the sampling process. Prior research in the realm of diffusion models has introduced the idea of classifier-guidance (Nichol et al., 2021) and classifier-free guidance (Ho & Salimans, 2022), where the predicted posterior mean is shifted with the gradient of either explicit or implicit classifier. However, these methods require labels as the source of guidance while sampling, which are unavailable during out-of-sample inference. We observe that the forward process of the diffusion model, which sequentially corrupts the data distribution to a standard normal distribution, intuitively aligns with the process of smoothing fine-grained data into a coarser-grained representation, both of which result in a gradual loss of finer distribution features. This provides the insights that intrinsic features within data granularities may also serve as a source of guidance.

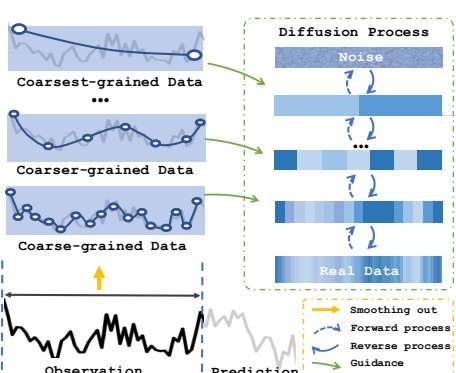

Figure 1: The process of smoothing data from finest-grained to coarsest-grained naturally aligns with the diffusion process.

In this paper, we propose a novel **M**ulti-**G**ranularity **T**ime **S**eries **D**iffusion (**MG-TSD**) model that leverages multiple granularity levels within data to guide the learning process of diffusion models. The coarse-grained data at different granularity levels are utilized as targets to guide the learning of the denoising process. These targets serve as constraints for the intermediate latent states, ensuring a regularized sampling path that preserves the trends and patterns within the coarse-grained data. They introduce inductive bias which promotes the generation of coarser features during intermediate steps and facilitates the recovery of finer features in subsequent diffusion steps. Consequently, this design reduces variability and results in high-quality predictions. Our key contributions can be summarized as below:

1. We introduce a novel **MG-TSD** model with an innovatively designed multi-granularity guidance loss function that efficiently guides the diffusion learning process, resulting in reliable sampling paths and more precise forecasting results.

2. We provide a concise implementation that leverages coarse-grained data instances at various granularity levels. Furthermore, we explore the optimal configuration for different granularity levels and propose a practical rule of thumb.

3. Extensive experiments conducted on real-world datasets demonstrate the superiority of the proposed model, achieving the best performance compared to the state-of-the-art methods.

## 2 BACKGROUND

### 2.1 DENOISING DIFFUSION PROBABILISTIC MODELS

Suppose $\boldsymbol{x}_0 \sim q_{\mathcal{X}}(\boldsymbol{x}_0)$ is a multivariate vector from space $\mathcal{X} = \mathbb{R}^D$. Denoising diffusion probabilistic models aim to learn a model distribution $p_\theta(\boldsymbol{x}_0)$ that approximates the data distribution $q(\boldsymbol{x}_0)$. Briefly, they are latent variable models of the form $p_\theta(\boldsymbol{x}_0) = \int p_\theta(\boldsymbol{x}_{0:N}) \mathrm{d}\boldsymbol{x}_{1:N}$, where $\boldsymbol{x}_n$ for $n = 1, \ldots, N$ is a sequence of latent variables in the same sample space as $\boldsymbol{x}_0$. The denoising diffusion models are composed of two processes: the forward process and the reverse process. During the forward process, a small amount of Gaussian noise is added gradually in $N$ steps to samples. It is characterized by the following Markov chain: $q(\boldsymbol{x}_{1:N}|\boldsymbol{x}_0) = \prod_{n=1}^{N} q(\boldsymbol{x}_n|\boldsymbol{x}_{n-1})$, where $q(\boldsymbol{x}_n|\boldsymbol{x}_{n-1}) := \mathcal{N}(\sqrt{1 - \beta_n}\boldsymbol{x}_{n-1}, \beta_n \boldsymbol{I})$. The step sizes are controlled by a variance schedule $\{\beta_n \in (0,1)\}_{n=1}^{N}$, where $n$ represents a diffusion step. A nice property of the above process is that one can sample

at any arbitrary diffusion step in a closed form, let $\alpha_n := 1 - \beta_n$ and $\bar{\alpha}_n = \prod_{i=1}^{n} \alpha_i$. It has been shown that $\boldsymbol{x}_n = \sqrt{\bar{\alpha}_n}\boldsymbol{x}_0 + \sqrt{1 - \bar{\alpha}_n}\boldsymbol{\epsilon}$. The reverse diffusion process is to recreate the real samples from a Gaussian noise input. It is defined as a Markov chain with learned Gaussian transitions starting with $p(\boldsymbol{x}_N) = \mathcal{N}(\boldsymbol{x}_N; \boldsymbol{0}, \boldsymbol{I})$. The reverse process is characterized as $p_\theta(\boldsymbol{x}_{0:N}) :=$ $p(\boldsymbol{x}_N) \prod_{n=N}^{1} p_\theta(\boldsymbol{x}_{n-1}|\boldsymbol{x}_n)$, where $p_\theta(\boldsymbol{x}_{n-1}|\boldsymbol{x}_n) := \mathcal{N}(\boldsymbol{x}_{n-1}; \mu_\theta(\boldsymbol{x}_n, n), \Sigma_\theta(\boldsymbol{x}_n, n)\boldsymbol{I})$; $\mu_\theta :$ $\mathbb{R}^D \times \mathbb{N} \to \mathbb{R}^D$ and $\Sigma_\theta : \mathbb{R}^D \times \mathbb{N} \to \mathbb{R}^+$ take the variable $\boldsymbol{x}_n \in \mathbb{R}^D$ and the diffusion step $n \in \mathbb{N}$ as inputs, and share the parameters $\theta$. The parameters in the model are optimized to minimize the negative log-likelihood $\min_\theta \mathbb{E}_{\boldsymbol{x}_0 \sim q(\boldsymbol{x}_0)}[-\log p_\theta(\boldsymbol{x}_0)]$ via a variational bound. According to denoising diffusion probabilistic models (DDPM) in Ho et al. (2020), the parameterization of $p_\theta(\boldsymbol{x}_{n-1}|\boldsymbol{x}_n)$ is chosen as:

$$\mu_\theta(\boldsymbol{x}_n, n) = \frac{1}{\sqrt{\alpha_n}}\left(\boldsymbol{x}_n - \frac{1 - \alpha_n}{\sqrt{1 - \bar{\alpha}_n}}\boldsymbol{\epsilon}_\theta(\sqrt{\bar{\alpha}_n}\boldsymbol{x}_0 + \sqrt{1 - \bar{\alpha}_n}\boldsymbol{\epsilon}, n)\right), \tag{1}$$

where $\boldsymbol{\epsilon}_\theta$ is a network which predicts $\boldsymbol{\epsilon} \sim \mathcal{N}(\boldsymbol{0}, \boldsymbol{I})$ from $\boldsymbol{x}_n$. We simplify the objective function into

$$L_n^{\text{simple}} = \mathbb{E}_{n, \boldsymbol{\epsilon}_n, \boldsymbol{x}_0}\left[\|\boldsymbol{\epsilon}_n - \boldsymbol{\epsilon}_\theta(\sqrt{\bar{\alpha}_n}\boldsymbol{x}_0 + \sqrt{1 - \bar{\alpha}_n}\boldsymbol{\epsilon}_n, n)\|^2\right]. \tag{2}$$

Once trained, we can iteratively sample from the reverse process $p_\theta(\boldsymbol{x}_{n-1}|\boldsymbol{x}_n)$ to reconstruct $\boldsymbol{x}_0$.

## 2.2 TimeGrad Model

We treat the time series forecasting task as a conditional generation task and utilize the diffusion models presented in Section 2.1 as the backbone generative model. TimeGrad model is a related work by Rasul et al. (2020) which first explored the use of diffusion models for forecasting multivariate time series. Consider a contiguous time series sampled from the complete history training data, indexed from 1 to $T$. This time series is partitioned into a context window of interval $[1, t_0)$ and a prediction interval $[t_0, T]$. TimeGrad utilizes diffusion models from Ho et al. (2020) to learn the conditional distribution of the future timesteps of the multivariate time series given their past. An RNN is employed to capture the temporal dependencies, and the time series sequence up to timestep $t$ is encoded in the updated hidden state $\mathbf{h}_t$. Mathematically, TimeGrad models $q_{\mathcal{X}}(\boldsymbol{x}_{t_0:T}|\boldsymbol{x}_{1:t_0-1}) = \prod_{t=t_0}^{T} q_{\mathcal{X}}(\boldsymbol{x}_t|\boldsymbol{x}_{1:t-1}) \approx \prod_{t=t_0}^{T} q_{\mathcal{X}}(\boldsymbol{x}_t|\mathbf{h}_{t-1})$, where $\boldsymbol{x}_t \in \mathbb{R}^D$ denotes the time series at timestep $t$ and $\mathbf{h}_t = \text{RNN}_\psi(\boldsymbol{x}_t, \mathbf{h}_{t-1})$. Each factor is learned via a shared conditional denoising diffusion model. In contrast to Ho et al. (2020), the hidden states $\mathbf{h}_{t-1}$ are taken as an additional input in the denoising network $\boldsymbol{\epsilon}_\theta(\boldsymbol{x}_t^n, \mathbf{h}_{t-1}, n)$, and the loss function for timestep $t$ and diffusion step $n$ is given by:

$$\mathbb{E}_{\boldsymbol{\epsilon}, \boldsymbol{x}_{0,t}, n}[\|\boldsymbol{\epsilon} - \boldsymbol{\epsilon}_\theta(\sqrt{\bar{\alpha}_n}\boldsymbol{x}_{0,t} + \sqrt{1 - \bar{\alpha}_n}\boldsymbol{\epsilon}, n, \mathbf{h}_{t-1})\|^2], \tag{3}$$

where the first subscript in $\boldsymbol{x}_{0,t}$ represents the index of the diffusion step, while $t$ denotes the timestep within the time series.

## 2.3 Problem formulation

In the time series prediction task, let $\boldsymbol{X}^{(1)}$ represent the original observed data. The time series data is denoted as $\boldsymbol{X}^{(1)} = [\boldsymbol{x}_1^1, \ldots, \boldsymbol{x}_t^1, \ldots, \boldsymbol{x}_T^1]$, where $t$ represents the timestep $t \in [1, T]$ and $\boldsymbol{x}_t \in \mathbb{R}^D$. Specifically, our task is to model the conditional distribution of future timesteps of the time series $[\boldsymbol{x}_{t_0}^1, \ldots, \boldsymbol{x}_T^1]$ given the fixed window of history context. Mathematically, the problem we consider can be formulated as follows:

$$q_{\mathcal{X}}\left(\boldsymbol{x}_{t_0:T}^1 | \{\boldsymbol{x}_{1:t_0-1}^1\}\right) = \prod_{t=t_0}^{T} q_{\mathcal{X}}\left(\boldsymbol{x}_t^1 | \{\boldsymbol{x}_{1:t-1}^1\}\right). \tag{4}$$

## 3 Method

In this section, we provide an overview of the **MG-TSD** model architecture in Section 3.1, followed by a detailed discussion of the novel guided diffusion process module in Section 3.2, including the derivation of the heuristic loss function and its implementation across various granularity levels.

### 3.1 MG-TSD MODEL ARCHITECTURE

The proposed methodology consists of three key modules, as depicted in Figure 2.

**Multi-granularity Data Generator** is responsible for generating multi-granularity data from observations. In this module, various coarse-grained time series are obtained by smoothing out the fine-grained data using historical sliding windows with different sizes. Suppose $f$ is a pre-defined smoothing (for example, average) function, and $s^g$ is the pre-defined sliding window size for granularity level $g$. Then $\boldsymbol{X}^{(g)} = f(\boldsymbol{X}^{(1)}, s^g)$. The sliding windows are non-overlapping and the obtained coarse-grained data for granularity $g$ are replicated $s^g$ times to align over the timeline $[1, T]$.

**Temporal Process Module** is designed to capture the temporal dynamics of the multi-granularity time series data. We utilize RNN architecture on each granularity level $g$ separately to encode the time series sequence up to a specific timestep $t$ and the encoded hidden states are denoted as $\mathbf{h}_t^g$. The RNN cell type is implemented as GRU in Chung et al. (2014).

**Guided Diffusion Process Module** is designed to generate stable time series predictions at each timestep $t$. We utilize multi-granularity data as given targets to guide the diffusion learning process. A detailed discussion of the module can be found in Section 3.2.

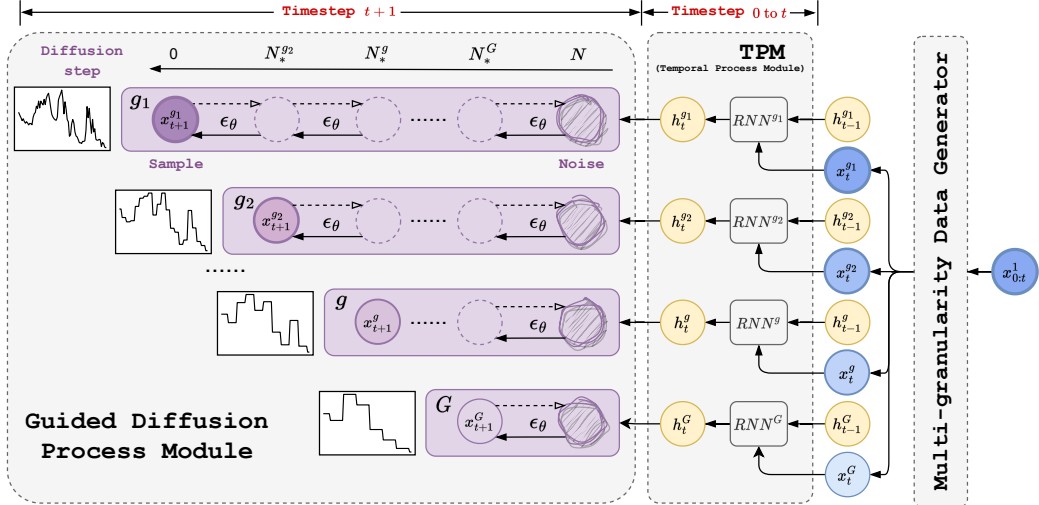

Figure 2: Overview of the Multi-Granularity Time Series Diffusion (**MG-TSD**) model, consisting of three key modules: **Multi-granularity Data Generator**, **Temporal Process Module (TPM)**, and **Guided Diffusion Process Module** for time series forecasting at a specific granularity level.

### 3.2 MULTI-GRANULARITY GUIDED DIFFUSION

In this section, we delve into the details of the **Guided Diffusion Process Module**, a key component in our model. Section 3.2.1 presents the derivation of a heuristic guidance loss for the two-granularity case. In Section 3.2.2, we generalize the loss to the multi-granularity case and provide a concise implementation to effectively utilize coarse-grained data across various granularity levels. Briefly, the optimization of the heuristic loss function can be simply achieved by training denoising diffusion models on the multi-granularity data with shared denoising network parameters and partially shared variance schedule.

### 3.2.1 COARSE-GRAINED GUIDANCE

Without loss of generality, consider two granularities: finest-grained data $\boldsymbol{x}_t^{g_1}$ ($g_1 = 1$) from $\boldsymbol{X}^{(g_1)}$ and coarse-grained data $\boldsymbol{x}_t^g$ from $\boldsymbol{X}^{(g)}$ at a fixed timestep $t$, where $1 < t < T$. We omit the subscript $t$ in the derivation for notation brevity. Suppose the denoising diffusion models presented in Section 2.1 are employed to approximate the distribution $q(\boldsymbol{x}^{g_1})$ and let the variance schedule be $\{\beta_n^1 = 1 - \alpha_n^1 \in (0, 1)\}_{n=1}^N$. Suppose $\boldsymbol{x}_0^{g_1} \sim q(\boldsymbol{x}_0^{g_1})$, where the subscript 0 denotes the index

of diffusion step. The diffusion models in Section 2.2 define a forward trajectory $q(\boldsymbol{x}_{0:N}^{g_1})$ and a $\theta$-parameterized reverse trajectory $p_\theta(\boldsymbol{x}_{0:N}^{g_1})$.

While Section 2.2 focuses on predicting samples over a specific timestep, it does not account for the intrinsic structure of time series, such as trends, which are represented by coarse-grained time series. In this paper, we guide the generation of samples by ensuring that the intermediate latent space retains the underlying time series structure. This is achieved by introducing coarse-grained targets $\boldsymbol{x}^g$ at intermediate diffusion step $N_*^g \in [1, N-1]$. Specifically, we establish the objective function as the log-likelihood of observed coarse-grained data $\boldsymbol{x}^g$ evaluated at the marginal distributions at diffusion step $N_*^g$, which can be expressed as $\log p_\theta(\boldsymbol{x}^g)$. With an appropriate choice of diffusion step $N_*^g$, the coarser features recovered from the denoising process could gain information from the realistic coarse-grained sample. Recall that the marginal distribution of latent variable at denoising step $N_*^g$ determined by the $\theta$-parameterized trajectory $p_\theta(\boldsymbol{x}_{N_*^g:N})$ can be expressed as:

$$p_\theta(\boldsymbol{x}_{N_*^g}) = \int p_\theta(\boldsymbol{x}_{N_*^g:N}) \mathrm{d}\boldsymbol{x}_{(N_*^g+1):N} = \int p(\boldsymbol{x}_N) \prod_{N_*^g+1}^{N} p_\theta(\boldsymbol{x}_{n-1}|\boldsymbol{x}_n) \mathrm{d}\boldsymbol{x}_{(N_*^g+1):N}, \quad (5)$$

where $\boldsymbol{x}_N \sim \mathcal{N}(\boldsymbol{0}, \boldsymbol{I})$, $p_\theta(\boldsymbol{x}_{n-1}|\boldsymbol{x}_n) = \mathcal{N}(\boldsymbol{x}_{n-1}; \boldsymbol{\mu}_\theta(\boldsymbol{x}_n, n), \boldsymbol{\Sigma}_\theta(\boldsymbol{x}_n, n))$.

To make the objective tractable, a common technique involves optimizing a variational lower bound on the likelihood in Equation 5. This can be achieved by specifying a latent variable sequence of length $N - N_*^g$, such that the joint distribution of $\boldsymbol{x}^g$ and these latent variables is available. Conveniently, we employ a diffusion process on $\boldsymbol{x}^g$ with a total of $N - N_*^g$ diffusion steps, defining a sequence of noisy samples $\boldsymbol{x}_{N_*^g+1}^g$, ..., $\boldsymbol{x}_N^g$ as realizations of the latent variable sequence. Then, the guidance objective can be expressed as:

$$\log p_\theta(\boldsymbol{x}^g) = \log \int p_\theta(\boldsymbol{x}_{N_*^g}^g, \boldsymbol{x}_{N_*^g+1}^g, \ldots, \boldsymbol{x}_N^g) \mathrm{d}\boldsymbol{x}_{(N_*^g+1):N}^g. \quad (6)$$

Applying the same technique as in Ho et al. (2020), the guidance objective function in Equation 6 simplifies the loss function of the diffusion models (see the Appendix A for proof details):

$$\mathbb{E}_{\boldsymbol{\epsilon}, \boldsymbol{x}^g, n}[\|\boldsymbol{\epsilon} - \boldsymbol{\epsilon}_\theta(\boldsymbol{x}_n^g, n)\|^2], \quad (7)$$

where $\boldsymbol{x}_n^g = (\prod_{i=N_*^g}^n \alpha_i^1)\boldsymbol{x}^g + \sqrt{1 - \prod_{i=N_*^g}^n \alpha_i^1}\boldsymbol{\epsilon}$ and $\boldsymbol{\epsilon} \sim \mathcal{N}(\boldsymbol{0}, \boldsymbol{I})$. When the variance schedule is chosen as $\{\alpha_n^1\}_{n=N_*^g}^N$, the loss function of the diffusion model in Ho et al. (2020) is equivalent to the guidance loss function presented in Equation 7.

### 3.2.2 MULTI-GRANULARITY GUIDANCE

In general, for $G$ granularity levels, data of different granularities generated by **Multi-granularity Data Generator** can be represented as $\boldsymbol{X}^{(1)}, \boldsymbol{X}^{(2)}, \ldots, \boldsymbol{X}^{(G)}$. We expect these coarse-grained data can guide the learning process of the diffusion model at different steps, serving as constraints along the sampling trajectory. For coarse-grained data at granularity level $g$, where $g \in \{2, \ldots, G\}$, we define the **share ratio** as $r_g := 1 - (N_*^g - 1)/N$. It represents the shared percentage of variance schedule between the $g$th granularity data and the finest-grained data. For the finest-grained data, $N_*^1 = 1$ and $r^1 = 1$. Formally, the variance schedule for granularity $g$ is defined as

$$\alpha_n^g(N_*^g) = \begin{cases} 1 & \text{if } n = 1, \ldots, N_*^g \\ \alpha_n^1 & \text{if } n = N_*^g + 1, \ldots, N, \end{cases} \quad (8)$$

and $\{\beta_n^g\}_{n=1}^N = \{1 - \alpha_n^g\}_{n=1}^N$. Accordingly, define $a_n^g(N_*^g) = \prod_{k=1}^n \alpha_k^g$, and $b_n^g(N_*^g) = 1 - a_n^g(N_*^g)$. We suppose $N_*^1 < N_*^2 \ldots < N_*^g < \ldots < N_*^G$, which represents the diffusion index for starting sharing the variance schedule across granularity level $g \in \{1, \ldots, G\}$. The starting index $N_*^g$ is larger for coarser granularity level, aligning with the intuition that the coarser-grained data loses fine distribution features to a greater extent and is expected to resemble the samples from earlier sampling steps.

Furthermore, we use the temporal hidden states for granularity level $g$ up to timestep $t$ from the **Temporal Process Module** as conditional inputs for the model to generate time series at corresponding granularity levels similar to Rasul et al. (2020). Then the guidance loss function $L^{(g)}(\theta)$

for $g$th-granularity $\boldsymbol{x}_{n,t}^g$ at timestep $t$ and diffusion step $n$, can be expressed as:

$$L^{(g)}(\theta) = \mathbb{E}_{\boldsymbol{\epsilon}, \boldsymbol{x}_{0,t}^g, n} \| (\boldsymbol{\epsilon} - \boldsymbol{\epsilon}_\theta(\sqrt{a_n^g}\boldsymbol{x}_{0,t}^g + \sqrt{b_n^g}\boldsymbol{\epsilon}, n, \mathbf{h}_{t-1}^g) \|_2^2, \tag{9}$$

where $\mathbf{h}_t^g = \mathrm{RNN}_\theta(\boldsymbol{x}_t^g, \mathbf{h}_{t-1}^g)$ is the updated hidden states from the last step.

The guidance loss function with $G - 1$ granularity levels of data is $L^{\mathrm{guidance}} = \sum_{g=2}^G \omega^g L^{(g)}(\theta)$, where $\omega^g \in [0, 1]$ is a hyper-parameter controlling the scale of guidance from granularity $g$.

**Training.** The training algorithm is in Algorithm 1. The final training objective is the weighted summation of loss for all granularities, including the finest granularity:

$$L^{\mathrm{final}} = \omega^1 L^{(1)}(\theta) + L^{\mathrm{guidance}}(\theta) = \sum_{g=1}^G \omega^g \mathbb{E}_{\boldsymbol{\epsilon}, \boldsymbol{x}_{0,t}^g, n}[\|\boldsymbol{\epsilon} - \boldsymbol{\epsilon}_\theta(\boldsymbol{x}_{n,t}^g, n, \mathbf{h}_{t-1}^g)\|^2], \tag{10}$$

where $\boldsymbol{x}_{n,t}^g = \sqrt{a_n^g}\boldsymbol{x}_{0,t}^g + \sqrt{b_n^g}\boldsymbol{\epsilon}$ and $\sum_{g=1}^G \omega^g = 1$. The denoising network parameters are shared across all granularities during training.

---

**Algorithm 1** Training procedure

---

**Input:** Context interval $[1, t_0)$; prediction interval $[t_0, T]$; number of diffusion step $N$; a set of share ratio for $g$ granularity (or equivalently $\{N_*^g, g \in \{1, \ldots, G\}\}$); generated multi-granularity data $[\boldsymbol{x}_1^g, \ldots, \boldsymbol{x}_{t_0}^g, \ldots, \boldsymbol{x}_T^g], g \in \{1, \ldots, G\}$; initial hidden states $\mathbf{h}_0^g, g \in \{1, \ldots, G\}$]

**repeat**
  1: Sample the multi-granularity time series $[\boldsymbol{x}_1^g, \ldots, \boldsymbol{x}_T^g], g \in \{1, \ldots, G\}$.
  2: Obtain $\mathbf{h}_t^g = \mathrm{RNN}^g(\boldsymbol{x}_t^g, \mathbf{h}_{t-1}^g), g \in \{1, \ldots, G\}, t \in [1, \ldots, T]$.
  3: **for** $t = t_0$ to $T$ **do**
  4:     Initialize $n \sim \mathrm{Uniform}(1, \ldots, N)$ and $\boldsymbol{\epsilon} \sim \mathcal{N}(\mathbf{0}, \boldsymbol{I})$
  5:     Reset the variance schedule $\{\beta_n^g = 1 - \alpha_n^g(N_*^g)\}_{n=1}^N, g \in \{1, \ldots, G\}$.
  6:     Compute loss $L^{\mathrm{final}}$ according to Equation 10
  7:     Take the gradient $\nabla_\theta L^{\mathrm{final}}$
  8: **end for**
**until** converged

---

**Inference.** Once the model is trained, our goal is to make predictions on the finest-grained data, up to a certain number of future prediction steps. Suppose that the last context window ends at timestep $t_0 - 1$, we use Algorithm 2 to perform the sampling procedure and generate a sample $\boldsymbol{x}_{t_0}^1$ for the next timestep. This process is repeated until reaching the desired forecast horizon. With different hidden states as conditional inputs, the model can sample time series at respective granularity levels.

---

**Algorithm 2** Inference procedure for each timestep $t \in [t_0, T]$

---

**Input:** Noise $\boldsymbol{x}_t^N \sim \mathcal{N}(\mathbf{0}, \boldsymbol{I})$ and hidden states $\mathbf{h}_{t-1}^g, g \in \{1, \ldots, G\}$

  1: **for** $n = N$ to $1$ **do**
  2:    **if** $n > 1$ **then**
  3:       Sample $\boldsymbol{z} \sim \mathcal{N}(\mathbf{0}, \boldsymbol{I})$
  4:    **else**
  5:       $\boldsymbol{z} = \mathbf{0}$
  6:    **end if**
  7:    **for** $g = 1$ to $G$ **do**
  8:       $\boldsymbol{x}_{n-1,t}^g = \frac{1}{\sqrt{\alpha_n^g}}(\boldsymbol{x}_{n,t}^g - \frac{\beta_n^g}{\sqrt{1-a_n^g}}\epsilon_\theta(\boldsymbol{x}_{n,t}^g, n, \mathbf{h}_{t-1}^g)) + \sqrt{\sigma_n^g}\boldsymbol{z}$, where $\sigma_n^g = \frac{1-a_{n-1}^g}{1-a_n^g}\beta_n^g$.
  9:    **end for**
10: **end for**
**Return:** $\boldsymbol{x}_{0,t}^g, g = 1$(finest-grained data); (Optional: $\boldsymbol{x}_{0,t}^g, g \in \{2, \ldots, G\}$)

---

**Selection of share ratio.** We propose a heuristic approach to help select the appropriate share ratio $r^g$, which is derived from $N_*^g$. We determine the choice of $N_*^g$ as the diffusion step at which the distance between two distributions $q(\boldsymbol{x}^g)$ and $p_\theta(\boldsymbol{x}_n^{g_1})$ is minimum, as shown below:

$$N_*^g := \arg\min_n \mathcal{D}(q(\boldsymbol{x}^g), p_\theta(\boldsymbol{x}_n^{g_1})), \tag{11}$$

where $\mathcal{D}$ is a measure for accessing discrepancy between two distributions, such as KL divergence. In practice, we first pre-train a TimeGrad model and then compute the $\text{CRPS}_{\text{sum}}$ between the coarse-grained targets and the samples along the sampling path of finest-grained data during inference. The range of steps where the $\text{CRPS}_{\text{sum}}$ values can consistently maintain relatively small values suggests a proper range of share ratios.

## 4 EXPERIMENTS

In this section, we conduct extensive experiments on six real-world datasets to evaluate the performance of the proposed **MG-TSD** model and compare it with previous state-of-the-art baselines.

### 4.1 SETTINGS

**Datasets.** We consider six real-word datasets characterized by a range of temporal dynamics, namely `Solar`, `Electricity`, `Traffic`, `Taxi`, `KDD-cup` and `Wikipedia`. The data is recorded at intervals of 30 minutes, 1 hour, or 1 day frequencies. Refer to Appendix C.1 for details.

**Evaluation Metrics.** We assess our model and all baselines using $\text{CRPS}_{\text{sum}}$ (Continuous Ranked Probability Score), a widely used metric for probabilistic time series forecasting, as well as $\text{NMAE}_{\text{sum}}$ (Normalized Mean Absolute Error) and $\text{NRMSE}_{\text{sum}}$ (Normalized Root Mean Squared Error). For detailed descriptions, refer to Appendix D.

**Baselines.** We assess the predictive performance of the proposed **MG-TSD** model in comparison with multivariate time series forecasting models, including Vec-LSTM-ind-scaling (Salinas et al., 2019), GP-scaling (Salinas et al., 2019), GP-Copula (Salinas et al., 2019), Transformer-MAF (Rasul et al., 2020), LSTM-MAF (Rasul et al., 2020), TimeGrad (Rasul et al., 2021), and TACTiS (Drouin et al., 2022). The MG-Input ensemble model serves as the baseline with multi-granularity inputs. It combines two TimeGrad models trained on one coarse-grained and finest-grained data respectively, and generates the final predictions by a weighted average of their outputs.

**Implementation details.** We train our model for 30 epochs using the Adam optimizer with a fixed learning rate of $10^{-5}$. We set the mini-batch size to 128 for solar and 32 for other datasets. The diffusion step is configured as 100. Additional hyper-parameters, such as share ratios, granularity levels, and loss weights, are detailed in Appendix C.3. All models are trained and tested on a single NVIDIA A100 80GB GPU.

### 4.2 RESULTS

The $\text{CRPS}_{\text{sum}}$ values averaged over 10 independent runs are reported in Table 1. The results show our model achieves the lowest $\text{CRPS}_{\text{sum}}$ and outperforms the baseline models across all six datasets. The MG-Input model exhibits marginal improvement on certain datasets when compared to the TimeGrad. This implies that while integrating multi-granularity information may result in some information gain, direct ensembling of coarse-grained outputs is inefficient in boosting performance.

### 4.3 ABLATION STUDY

**Share ratio of variance schedule.** To investigate the effect of share ratio, we evaluate the performance of **MG-TSD** using various share ratios across different coarse granularities. The experiment is conducted in a two-granularity setting, where one coarse granularity is utilized to guide the learning process for the finest-grained data. Table 2 shows that for each coarse granularity level, the $\text{CRPS}_{\text{sum}}$ values initially decrease to their lowest values and then ascend again as the share ratio gets larger. Furthermore, we observe for coarser granularities, the model performs better with a smaller share ratio. This suggests that the model achieves optimal performance when the share ratio is chosen at the step where the coarse-grained samples most closely resemble intermediate states. Utilizing 4-hour or 6-hour granularity as guidance greatly enhances the model performance. However, the improvement in performance diminishes as the granularity becomes coarser, such as 12 hours or 24 hours, possibly due to the greater loss of information on local fluctuations.

In practice, the selection of share ratio can follow the heuristic rule outlined in Section 3.2.2. Figure 3 provides illustrative plots for the share ratio selection curve of different granularities. The blue

Table 1: Comparison of CRPS$_{sum}$ (smaller is better) of models on six real-world datasets. The reported mean and standard error are obtained from 10 re-training and evaluation independent runs.

| Method | Solar | Electricity | Traffic | KDD-cup | Taxi | Wikipedia |
|---|---|---|---|---|---|---|
| Vec-LSTM ind-scaling | $0.4825_{\pm 0.0027}$ | $0.0949_{\pm 0.0175}$ | $0.0915_{\pm 0.0197}$ | $0.3560_{\pm 0.1667}$ | $0.4794_{\pm 0.0343}$ | $0.1254_{\pm 0.0174}$ |
| GP-Scaling | $0.3802_{\pm 0.0052}$ | $0.0499_{\pm 0.0031}$ | $0.0753_{\pm 0.0152}$ | $0.2983_{\pm 0.0448}$ | $0.2265_{\pm 0.0210}$ | $0.1351_{\pm 0.0612}$ |
| GP-Copula | $0.3612_{\pm 0.0035}$ | $0.0287_{\pm 0.0005}$ | $0.0618_{\pm 0.0018}$ | $0.3157_{\pm 0.0462}$ | $0.1894_{\pm 0.0087}$ | $0.0669_{\pm 0.0009}$ |
| LSTM-MAF | $0.3427_{\pm 0.0082}$ | $0.0312_{\pm 0.0046}$ | $0.0526_{\pm 0.0021}$ | $0.2919_{\pm 0.1486}$ | $0.2295_{\pm 0.0082}$ | $0.0763_{\pm 0.0051}$ |
| Transformer-MAF | $0.3552_{\pm 0.0053}$ | $0.0272_{\pm 0.0017}$ | $0.0499_{\pm 0.0011}$ | $0.2951_{\pm 0.0504}$ | $0.1531_{\pm 0.0038}$ | $0.0644_{\pm 0.0037}$ |
| TimeGrad | $0.3335_{\pm 0.0653}$ | $0.0232_{\pm 0.0035}$ | $0.0414_{\pm 0.0112}$ | $0.2902_{\pm 0.2178}$ | $0.1255_{\pm 0.0207}$ | $0.0555_{\pm 0.0088}$ |
| TACTiS | $0.4209_{\pm 0.0330}$ | $0.0259_{\pm 0.0019}$ | $0.1093_{\pm 0.0076}$ | $0.5406_{\pm 0.1584}$ | $0.2070_{\pm 0.0159}$ | — |
| MG-Input | $0.3239_{\pm 0.0427}$ | $0.0238_{\pm 0.0035}$ | $0.0658_{\pm 0.0065}$ | $0.2977_{\pm 0.1163}$ | $0.1592_{\pm 0.0087}$ | $0.0567_{\pm 0.0091}$ |
| MG-TSD | $\mathbf{0.3081_{\pm 0.0099}}$ | $\mathbf{0.0149_{\pm 0.0017}}$ | $\mathbf{0.0323_{\pm 0.0125}}$ | $\mathbf{0.1837_{\pm 0.0865}}$ | $\mathbf{0.1159_{\pm 0.0132}}$ | $\mathbf{0.0529_{\pm 0.0054}}$ |

Table 2: Influence of share ratios for different granularities on `Solar` dataset. The reported mean and standard error are obtained from 10 re-training and evaluation independent runs.

| Ratio | 4 hour | | | 6 hour | | |
|---|---|---|---|---|---|---|
| | CRPS$_{sum}$ | NMAE$_{sum}$ | NRMSE$_{sum}$ | CRPS$_{sum}$ | NMAE$_{sum}$ | NRMSE$_{sum}$ |
| 20% | $0.3489_{\pm 0.0190}$ | $0.3826_{\pm 0.0200}$ | $0.7177_{\pm 0.0445}$ | $0.3378_{\pm 0.0305}$ | $0.3703_{\pm 0.0368}$ | $0.6916_{\pm 0.0536}$ |
| 40% | $0.3405_{\pm 0.0415}$ | $0.3792_{\pm 0.0386}$ | $0.6870_{\pm 0.0870}$ | $0.3275_{\pm 0.0250}$ | $0.3608_{\pm 0.0267}$ | $0.6650_{\pm 0.0374}$ |
| 60% | $0.3268_{\pm 0.0475}$ | $0.3604_{\pm 0.0463}$ | $0.6579_{\pm 0.0919}$ | $\mathbf{0.3166_{\pm 0.0376}}$ | $\mathbf{0.3491_{\pm 0.0368}}$ | $\mathbf{0.6478_{\pm 0.0696}}$ |
| 80% | $\mathbf{0.3172_{\pm 0.0249}}$ | $\mathbf{0.3510_{\pm 0.0240}}$ | $\mathbf{0.6515_{\pm 0.051}}$ | $0.3221_{\pm 0.0425}$ | $0.3555_{\pm 0.0443}$ | $0.6542_{\pm 0.0747}$ |
| 100% | $0.3178_{\pm 0.0342}$ | $0.3480_{\pm 0.0356}$ | $0.6591_{\pm 0.0503}$ | $0.3232_{\pm 0.0396}$ | $0.3548_{\pm 0.0417}$ | $0.6550_{\pm 0.0660}$ |

| Ratio | 12 hour | | | 24 hour | | |
|---|---|---|---|---|---|---|
| | CRPS$_{sum}$ | NMAE$_{sum}$ | NRMSE$_{sum}$ | CRPS$_{sum}$ | NMAE$_{sum}$ | NRMSE$_{sum}$ |
| 20% | $0.3440_{\pm 0.0391}$ | $0.3767_{\pm 0.0450}$ | $0.6999_{\pm 0.0772}$ | $0.3315_{\pm 0.0266}$ | $0.3693_{\pm 0.0298}$ | $0.6801_{\pm 0.0554}$ |
| 40% | $0.3374_{\pm 0.0370}$ | $0.3713_{\pm 0.0346}$ | $0.6837_{\pm 0.0641}$ | $\mathbf{0.3276_{\pm 0.0358}}$ | $\mathbf{0.3612_{\pm 0.0361}}$ | $\mathbf{0.6722_{\pm 0.0552}}$ |
| 60% | $\mathbf{0.3240_{\pm 0.0382}}$ | $\mathbf{0.3597_{\pm 0.0388}}$ | $\mathbf{0.6694_{\pm 0.0746}}$ | $0.3382_{\pm 0.0343}$ | $0.3737_{\pm 0.0365}$ | $0.6878_{\pm 0.0655}$ |
| 80% | $0.3391_{\pm 0.0390}$ | $0.3719_{\pm 0.0403}$ | $0.6953_{\pm 0.0691}$ | $0.3288_{\pm 0.0460}$ | $0.3639_{\pm 0.0476}$ | $0.6741_{\pm 0.0929}$ |
| 100% | $0.3284_{\pm 0.0323}$ | $0.3538_{\pm 0.0450}$ | $0.6609_{\pm 0.0917}$ | $0.3407_{\pm 0.0248}$ | $0.3692_{\pm 0.0244}$ | $0.6933_{\pm 0.0528}$ |

curve in each plot represents CRPS$_{sum}$ values between coarse-grained targets and 1-hour samples come from 1-gran(finest-gran) model at each intermediate denoising step; each point on the orange polylines represents the CRPS$_{sum}$ value of 1-hour predictions by 2-gran MG-TSD models with different share ratios ranging from [0.2, 0.4, 0.6, 0.8, 1.0], and the lowest point of the line segment can be used to characterize the most suitable share ratio for the corresponding granularity.

The diffusion steps that can achieve relatively small CRPS$_{sum}$ values are colored in grey, suggesting a proper range for the share ratio at which the model can achieve satisfactory performance. From the plots, a strong correlation exists between the polyline of CRPS$_{sum}$ calculated during test time and the share ratio selection curve, which validates the effectiveness of the selection rule. In addition, as granularity transitions from fine to coarse (4h→6h→12h→24h), the diffusion steps at which the distribution most resembles the coarse-grained targets increase (approximately at steps 20→40→60→60). This comparison shows the similarity between the diffusion process and the smoothing process from the finest-grained to coarse-grained data, both of which involve a gradual loss of finer characteristics from the finest-grained data through a smooth and convex transformation.

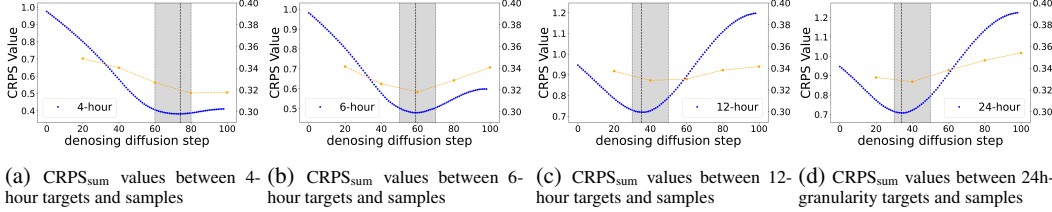

(a) CRPS$_{sum}$ values between 4-hour targets and samples (b) CRPS$_{sum}$ values between 6-hour targets and samples (c) CRPS$_{sum}$ values between 12-hour targets and samples (d) CRPS$_{sum}$ values between 24h-granularity targets and samples

Figure 3: Selection of share ratio for **MG-TSD** models

**The number of granularities.** We further explore the impact of the number of granularities on the **MG-TSD** model. As presented in Table 3, increasing the number of granularity levels generally boosts the performance of the **MG-TSD** model, which demonstrates that the introduction of

multi-granularity information effectively guides the learning process of diffusion models. However, the marginal benefit diminishes with the increase in granularity amounts. The results suggest that utilizing four to five granularity levels generally suffices for achieving favorable performance.

Table 3: Influence of the number of granularities on **MG-TSD** performance for `Solar` and `Electricity` Dataset.

| Num of gran | Solar | | | Electricity | | |
|---|---|---|---|---|---|---|
| | **CRPS**$_{\text{sum}}$ | **NMAE**$_{\text{sum}}$ | **NRMSE**$_{\text{sum}}$ | **CRPS**$_{\text{sum}}$ | **NMAE**$_{\text{sum}}$ | **NRMSE**$_{\text{sum}}$ |
| 2 | $0.3172_{\pm0.0249}$ | $0.3510_{\pm0.0240}$ | $0.6515_{\pm0.0571}$ | $0.0174_{\pm0.0042}$ | $0.0226_{\pm0.0071}$ | $0.0296_{\pm0.0086}$ |
| 3 | $0.3110_{\pm0.0329}$ | $0.3494_{\pm0.0378}$ | $0.6452_{\pm0.0632}$ | $0.0160_{\pm0.0020}$ | $0.0198_{\pm0.0029}$ | $0.0262_{\pm0.0039}$ |
| 4 | $\mathbf{0.3081}_{\pm0.0099}$ | $\mathbf{0.3445}_{\pm0.0102}$ | $\mathbf{0.6245}_{\pm0.0268}$ | $\mathbf{0.0149}_{\pm0.0017}$ | $\mathbf{0.0178}_{\pm0.0018}$ | $\mathbf{0.0241}_{\pm0.0030}$ |
| 5 | $0.3093_{\pm0.0411}$ | $0.3430_{\pm0.0451}$ | $0.6117_{\pm0.0746}$ | $0.0153_{\pm0.0027}$ | $0.0181_{\pm0.0043}$ | $0.0254_{\pm0.0058}$ |
| Num of gran | Traffic | | | KDD-cup | | |
| | **CRPS**$_{\text{sum}}$ | **NMAE**$_{\text{sum}}$ | **NRMSE**$_{\text{sum}}$ | **CRPS**$_{\text{sum}}$ | **NMAE**$_{\text{sum}}$ | **NRMSE**$_{\text{sum}}$ |
| 2 | $0.0347_{\pm0.0020}$ | $0.0396_{\pm0.0022}$ | $0.0593_{\pm0.0043}$ | $0.2427_{\pm0.1167}$ | $0.3171_{\pm0.1557}$ | $0.3745_{\pm0.1652}$ |
| 3 | $0.0334_{\pm0.0034}$ | $0.0382_{\pm0.0035}$ | $0.0574_{\pm0.0066}$ | $0.2414_{\pm0.1619}$ | $0.3030_{\pm0.1789}$ | $0.3808_{\pm0.2168}$ |
| 4 | $0.0326_{\pm0.0041}$ | $0.0374_{\pm0.0048}$ | $0.0573_{\pm0.0050}$ | $0.2198_{\pm0.1162}$ | $0.2893_{\pm0.1554}$ | $0.3315_{\pm0.1882}$ |
| 5 | $\mathbf{0.0323}_{\pm0.0125}$ | $\mathbf{0.0370}_{\pm0.0140}$ | $\mathbf{0.0563}_{\pm0.0230}$ | $\mathbf{0.1837}_{\pm0.0636}$ | $\mathbf{0.2463}_{\pm0.0865}$ | $\mathbf{0.3001}_{\pm0.0997}$ |

## 4.4 CASE STUDY

To illustrate the guidance effect of coarse-grained data, we visualize the ground truth and the predicted mean for both 1-hour and 4-hour granularity time series across four dimensions in the `Solar` dataset in Figure 4. For comparison, the prediction results for the 1-hour data from TimeGrad are also included. The results indicate that the TimeGrad model struggles to accurately capture the peaks in the series and tends to underestimate the peaks in solar energy. In the **MG-TSD** model, the coarse-grained samples display a more robust capacity to capture the trends, subsequently guiding the generation of more precise fine-grained data.

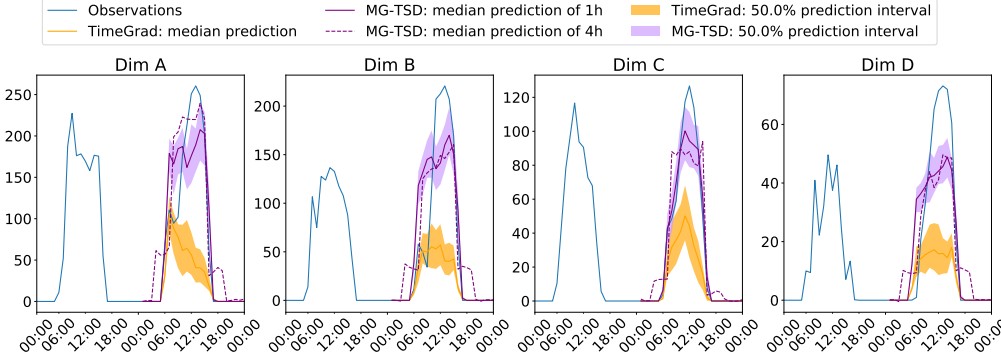

Figure 4: Visualization of the ground-truth (`Solar` dataset), **MG-TSD** predicted mean for 4-hour and 1-hour time series, and TimeGrad predicted mean for the 1-hour time series. Additionally, the 50% prediction intervals for the 1-hour data are also included. These plots represent some illustrative dimensions out of 370 dimensions from the first 24-hour rolling-window.

## 5 CONCLUSION

In this paper, we introduce a novel **M**ulti-**G**ranularity **T**ime **S**eries **D**iffusion (**MG-TSD**) model, which leverages the inherent granularity levels within the data as given targets at intermediate diffusion steps to guide the learning process of diffusion models. We derive a novel multi-granularity guidance diffusion loss function and propose a concise implementation method to effectively utilize coarse-grained data across various granularity levels. Extensive experiments conducted on real-world datasets demonstrate that **MG-TSD** outperforms existing time series prediction methods.

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

## A  APPENDIX: DERIVATION OF LOSS FUNCTION

Recall that we specify a sequence of noisy samples $\boldsymbol{x}^g_{N^g_*+1}, \ldots, \boldsymbol{x}^g_N$ by applying the forward process on $\boldsymbol{x}^g$. The superscript in $N^g_*$ is suppressed for notation brevity. Suppose the coarse-grained data $\boldsymbol{x}^g_{N_*} \sim q(\boldsymbol{x}^g)$, where the subscript notation $N_*$ indicates that the observed $\boldsymbol{x}^g$ is treated as a sample from the distribution. (In the diffusion model, the subscript is typically denoted as 0, but we start with $N_*$ to simplify the derivation).

$$
\begin{aligned}
\log p_\theta(\boldsymbol{x}^g_{N_*}) &\leq -\log p_\theta(\boldsymbol{x}^g_{N_*}) + D_{\mathrm{KL}}(q(\boldsymbol{x}^g_{(N_*+1):N}|\boldsymbol{x}^g_{N_*})\|p_\theta(\boldsymbol{x}^g_{(N_*+1):N}|\boldsymbol{x}^g_{N_*})) \\
&= -\log p_\theta(\boldsymbol{x}^g_{N_*}) + \mathbb{E}_{\boldsymbol{x}^g_{(N_*+1):N} \sim q(\boldsymbol{x}^g_{(N_*+1):N}|\boldsymbol{x}^g_{N_*})}\left[\log \frac{q(\boldsymbol{x}^g_{(N_*+1):N}|\boldsymbol{x}^g_{N_*})}{p_\theta(\boldsymbol{x}^g_{N_*:N})/p_\theta(\boldsymbol{x}^g_{N_*})}\right] \\
&= -\log p_\theta(\boldsymbol{x}^g_{N_*}) + \mathbb{E}_q\left[\log \frac{q(\boldsymbol{x}^g_{(N_*+1):N}|\boldsymbol{x}^g_{N_*})}{p_\theta(\boldsymbol{x}^g_{N_*:N})} + \log p_\theta(\boldsymbol{x}^g_{N_*})\right] \\
&= \mathbb{E}_q\left[\log \frac{q(\boldsymbol{x}^g_{(N_*+1):N}|\boldsymbol{x}^g_{N_*})}{p_\theta(\boldsymbol{x}^g_{N_*:N})}\right]
\end{aligned}
\tag{12}
$$

Then, the training objective can be performed by optimizing the usual variational lower bound shown below:

$$
L_{\mathrm{VLB}} = \mathbb{E}_{q(\boldsymbol{x}^g_{N_*:N})}\left[\log \frac{q(\boldsymbol{x}^g_{(N_*+1):N}|\boldsymbol{x}^g_{N_*})}{p_\theta(\boldsymbol{x}^g_{N_*:N})}\right] \geq -\mathbb{E}_{q(\boldsymbol{x}^g_{N_*})} \log p_\theta(\boldsymbol{x}^g_{N_*})
\tag{13}
$$

It is obvious that the objective $L_{\mathrm{VLB}}$ is equivalent to the that of diffusion model in Ho et al. (2020) when employing diffusion models on $\boldsymbol{x}^g$ with $N - N_*$ steps. The forward process is defined as $q(\boldsymbol{x}^g_{(N_*+1):N}|\boldsymbol{x}^g_{N_*}) = \prod_{n=N_*}^N q(\boldsymbol{x}^g_n|\boldsymbol{x}^g_{n-1})$, where $q(\boldsymbol{x}^g_n|\boldsymbol{x}^g_{n-1}) := \mathcal{N}(\sqrt{1-\beta^g_n}\boldsymbol{x}^g_{n-1}, \beta^g_n \boldsymbol{I})$. The $\{\beta^g_n\}^N_{n=N_*}$ share values with the variance schedule $\{\beta^1_n\}^N_{n=1}$ of the finest-grained data from index $N_*$. And, the reverse process is defined by the $\theta$-parameterized trajectory. Then following the same technique in Ho et al. (2020), the $L_{\mathrm{VLB}}$ can reduce to the usual loss of diffusion models.

## B  APPENDIX: EXPERIMENTS

### B.1  BENCHMARK EXPERIMENTS

The results of the benchmark experiments, evaluated based on the metrics NRMSE$_{\mathrm{sum}}$ and NMAE$_{\mathrm{sum}}$, are presented in Table 4 and Table 5 respectively. In the experiments, we include four extra baseline models for a more comprehensive comparison: TimeDiff (Shen & Kwok, 2023), D$^3$VAE (Li et al., 2022), PatchTST (Nie et al., 2022), and AutoFormer (Wu et al., 2021).

Table 4: Comparison of NRMSE$_{\mathrm{sum}}$ (smaller is better) of models on six real-world datasets. The reported mean and standard error are obtained from 10 re-training and evaluation independent runs.

| Method | Solar | Electricity | Traffic | KDD-cup | Taxi | Wikipedia |
|---|---|---|---|---|---|---|
| Vec-LSTM ind-scaling | $0.9952_{\pm 0.0077}$ | $0.1439_{\pm 0.0228}$ | $0.1451_{\pm 0.0248}$ | $0.4461_{\pm 0.1833}$ | $0.6398_{\pm 0.0390}$ | $0.1618_{\pm 0.0162}$ |
| GP-Scaling | $0.9004_{\pm 0.0095}$ | $0.0811_{\pm 0.0062}$ | $0.1469_{\pm 0.0181}$ | $0.3445_{\pm 0.0621}$ | $0.3598_{\pm 0.0285}$ | $0.1710_{\pm 0.1006}$ |
| GP-Copula | $0.8279_{\pm 0.0053}$ | $0.0512_{\pm 0.0009}$ | $0.1282_{\pm 0.0033}$ | $\mathbf{0.2605}_{\pm 0.0227}$ | $0.3125_{\pm 0.0113}$ | $0.0930_{\pm 0.0076}$ |
| Autoformer | $0.7046_{\pm 0.0000}$ | $0.0475_{\pm 0.0000}$ | $0.0951_{\pm 0.0000}$ | $0.8984_{\pm 0.0000}$ | $0.3498_{\pm 0.0000}$ | $0.1052_{\pm 0.0000}$ |
| PatchTST | $0.7270_{\pm 0.0000}$ | $0.0474_{\pm 0.0000}$ | $0.1897_{\pm 0.0000}$ | $0.5137_{\pm 0.0000}$ | $0.3690_{\pm 0.0000}$ | $0.0915_{\pm 0.0000}$ |
| D$^3$VAE | $0.7472_{\pm 0.0508}$ | $0.1640_{\pm 0.0928}$ | $0.4722_{\pm 0.1197}$ | $0.5628_{\pm 0.0419}$ | $0.7624_{\pm 0.5598}$ | $2.2094_{\pm 2.1646}$ |
| TimeDiff | $1.5985_{\pm 0.0359}$ | $0.3714_{\pm 0.0073}$ | $0.5520_{\pm 0.0087}$ | $0.4955_{\pm 0.0147}$ | $0.5479_{\pm 0.0084}$ | $0.1412_{\pm 0.0099}$ |
| TimeGrad | $0.6953_{\pm 0.0845}$ | $0.0348_{\pm 0.0057}$ | $0.0653_{\pm 0.0244}$ | $0.4092_{\pm 0.1332}$ | $0.2365_{\pm 0.0386}$ | $0.0870_{\pm 0.0106}$ |
| TACTiS | $0.8532_{\pm 0.0851}$ | $0.0427_{\pm 0.0023}$ | $0.2270_{\pm 0.0159}$ | $0.6513_{\pm 0.1767}$ | $0.3387_{\pm 0.0097}$ | - |
| MG-TSD | $\mathbf{0.6178}_{\pm 0.0418}$ | $\mathbf{0.0241}_{\pm 0.0030}$ | $\mathbf{0.0563}_{\pm 0.0230}$ | $0.3001_{\pm 0.0997}$ | $\mathbf{0.2334}_{\pm 0.0313}$ | $\mathbf{0.0810}_{\pm 0.0057}$ |

Table 5: Comparison of NMAE$_{sum}$ (smaller is better) of models on six real-world datasets. The reported mean and standard error are obtained from 10 re-training and evaluation independent runs.

| Method | Solar | Electricity | Traffic | KDD-cup | Taxi | Wikipedia |
|---|---|---|---|---|---|---|
| Vec-LSTM ind-scaling | $0.5091_{\pm 0.0027}$ | $0.1261_{\pm 0.0211}$ | $0.1042_{\pm 0.0228}$ | $0.4193_{\pm 0.1902}$ | $0.4974_{\pm 0.0351}$ | $0.1416_{\pm 0.0180}$ |
| GP-Scaling | $0.4945_{\pm 0.0065}$ | $0.0648_{\pm 0.0046}$ | $0.0975_{\pm 0.0163}$ | $0.2892_{\pm 0.0550}$ | $0.2867_{\pm 0.0264}$ | $0.1452_{\pm 0.1029}$ |
| GP-Copula | $0.4302_{\pm 0.0046}$ | $0.0312_{\pm 0.0007}$ | $0.0769_{\pm 0.0022}$ | $\mathbf{0.2140}_{\pm 0.0124}$ | $0.2390_{\pm 0.0098}$ | $0.0659_{\pm 0.0061}$ |
| Autoformer | $0.6368_{\pm 0.0000}$ | $0.0388_{\pm 0.0000}$ | $0.0684_{\pm 0.0000}$ | $0.7658_{\pm 0.0000}$ | $0.2652_{\pm 0.0000}$ | $0.1239_{\pm 0.0000}$ |
| PatchTST | $0.4351_{\pm 0.0000}$ | $0.0350_{\pm 0.0000}$ | $0.1219_{\pm 0.0000}$ | $0.4497_{\pm 0.0000}$ | $0.2887_{\pm 0.0000}$ | $0.0625_{\pm 0.0000}$ |
| D$^3$VAE | $0.4457_{\pm 0.0377}$ | $0.1434_{\pm 0.0892}$ | $0.3992_{\pm 0.1177}$ | $0.4874_{\pm 0.0520}$ | $0.6080_{\pm 0.5061}$ | $2.0151_{\pm 2.0005}$ |
| TimeDiff | $1.3343_{\pm 0.0305}$ | $0.3519_{\pm 0.0075}$ | $0.4782_{\pm 0.0058}$ | $0.3630_{\pm 0.0127}$ | $0.4521_{\pm 0.0102}$ | $0.1146_{\pm 0.0106}$ |
| TimeGrad | $0.3694_{\pm 0.0400}$ | $0.0266_{\pm 0.0049}$ | $0.0410_{\pm 0.0089}$ | $0.3614_{\pm 0.1334}$ | $0.1365_{\pm 0.0193}$ | $0.0631_{\pm 0.008}$ |
| TACTiS | $0.4448_{\pm 0.0313}$ | $0.0310_{\pm 0.0015}$ | $0.1352_{\pm 0.0159}$ | $0.6078_{\pm 0.1718}$ | $0.2244_{\pm 0.0036}$ | - |
| MG-TSD | $\mathbf{0.3347}_{\pm \mathbf{0.0220}}$ | $\mathbf{0.0178}_{\pm \mathbf{0.0018}}$ | $\mathbf{0.0370}_{\pm \mathbf{0.0140}}$ | $0.2463_{\pm 0.0865}$ | $\mathbf{0.1300}_{\pm \mathbf{0.0150}}$ | $\mathbf{0.0601}_{\pm \mathbf{0.0057}}$ |

## B.2 MORE EXPERIMENT SETTINGS

### B.2.1 PERFORMANCE FOR LONG-TERM FORECASTING

To evaluate the performance of MG-TSD for long-term forecasting, we maintain a fixed context length of 24 and extend the prediction length to 24, 48, 96, and 144. The results of the datasets `Solar` and `Eelectrity` are displayed in Figure 5.

The results in Figure 5 indicate that MG-TSD performs well for long-time forecasting. The results indicate that as the prediction length increases, the performance of our proposed method stays robust, exhibiting no sudden decline. Furthermore, our method consistently outperforms the competitive baseline. This performance advantage is anticipated to persist in future trends, with no indication of convergence between the approaches.

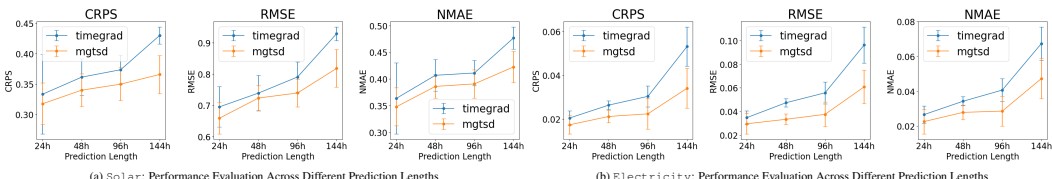

(a) `Solar`: Performance Evaluation Across Different Prediction Lengths
(b) `Electricity`: Performance Evaluation Across Different Prediction Lengths

Figure 5: Performance evaluation across different prediction horizons for MG-TSD with TimeGrad as the baseline Model. The context length is fixed at 24h and the prediction length is tested at 24h, 48h, 96h, and 144h. The average CRPS, NRMSE, and NMAE metrics are computed for both MG-TSD and the baseline over 10 independent runs, with error bars indicating the corresponding standard deviations.

### B.2.2 TIME AND MEMORY USAGE OF THE MG-TSD MODEL DURING TRAINING

Experiments have been conducted to evaluate the time and memory usage of the MG-TSD model during training across various granularities. These experiments were executed using a single A6000 card with 48G memory capacity. The Solar dataset was utilized in this context, with a batch size of 128, an input size of 552, 100 diffusion steps, and 30 epochs.

As illustrated in Figure 6, there is a linear increase in memory consumption with an increase in granularity. A slight surge in training time is also observed. These findings are coherent with the architecture of our model. In particular, each additional granularity results in the introduction of an extra RNN in the Temporal Process Module and an increase in computation within the Guided Diffusion Process Module. As per theoretical expectations, these resource consumptions should exhibit linear growth. The slight increase in training time can be ascribed to the design of the Multi-granularity Data Generator Module which enables parallel forward processes across different granularities, thus promoting acceleration. Moreover, it is pertinent to mention that an excessive increase in granularity may not notably boost the final prediction results, hence the granularity will be kept within a certain range. Therefore, the consumption of memory will not rise indefinitely.

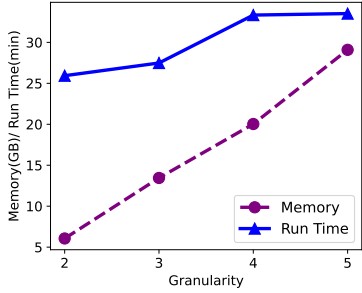

Figure 6: Comparison of Time and Memory Consumption at Different Granularity Levels in MG-TSD Model Training

### B.2.3 VARIATIONS IN THE FREQUENCY DOMAIN OF TIME SERIES DATA: THE IMPACT OF GRANULARITY AND DENOISING STEPS

We sampled series from Solar dateset and we conducted a Fast Fourier Transform to extract the seasonality components of the series, as well as the samples of different granularities and corresponding noisy samples along the forward diffusion process.

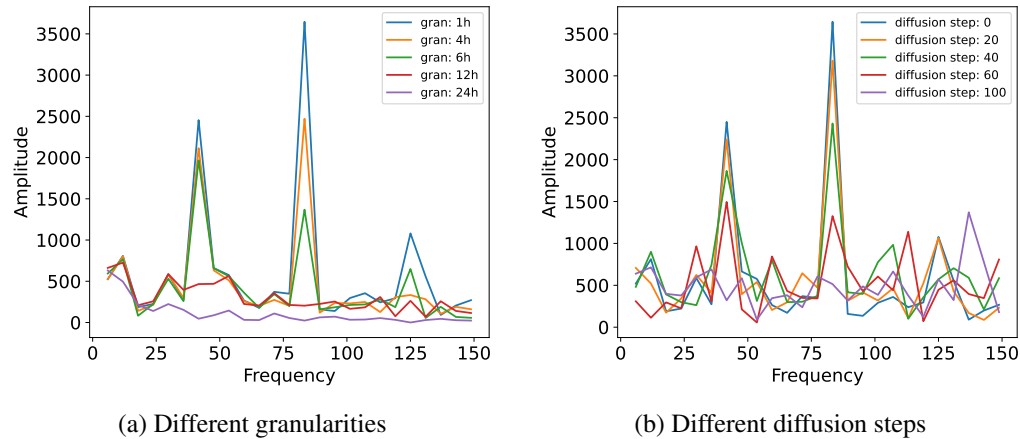

(a) Different granularities
(b) Different diffusion steps

Figure 7: Variations in the frequency domain of time series data: the impact of granularity and denoising steps.

As depicted in Figure 7(a), as granularity becomes coarser, the components of all outstanding frequencies get lower, while the high-frequency peak (around 125 and 80) diminishes quicker than lower-frequency peak (around 45). Figure 7(b) demonstrates the distribution of frequency components of the same noisy series with gradually ascending forward diffusion steps and the same pattern is observable. This empirical study indicates the connection between the forward diffusion process and the smoothing process from fine-grained data to coarse-grained data, both of which result in losing finer informative features.

## C APPENDIX: IMPLEMENTATION DETAILS

### C.1 BENCHMARK DATASETS

For our experiments, we use Solar, Electricity, Traffic, Taxi, KDD-cup and Wikipedia open-source datasets, with their properties listed in Table 6.

The dataset can be obtained through the links below.

(i) Solar: https://www.nrel.gov/grid/solar-power-data.html

(ii) `Electricity`: `https://archive.ics.uci.edu/dataset/321/electricityloaddiagrams20112014`

(iii) `Traffic`: `https://archive.ics.uci.edu/dataset/204/pems+sf`

(iv) `Taxi`: `https://www.nyc.gov/site/tlc/about/tlc-trip-record-data.page`

(v) `KDD-cup`: `https://www.kdd.org/kdd2018/kdd-cup`

(vi) `Wikipedia`: `https://github.com/mbohlkeschneider/gluon-ts/tree/mv_release/datasets`

| Name | Frequency | Number of series | Context length | Prediction length | Multi-granularity dictionary |
|------|-----------|------------------|----------------|-------------------|------------------------------|
| Solar | 1 hour | 137 | 24 | 24 | [1 hour, 4 hour, 12 hour, 24hour, 48 hour] |
| Electricity | 1 hour | 370 | 24 | 24 | [1 hour, 4 hour, 12 hour, 24 hour, 48 hour] |
| Traffic | 1 hour | 963 | 24 | 24 | [1 hour, 4 hour, 12 hour, 24 hour, 48 hour] |
| Taxi | 30 min | 1214 | 24 | 24 | [30 min , 2 hour, 6 hour, 12 hour, 24 hour] |
| KDD-cup | 1 hour | 270 | 48 | 48 | [1 hour, 4 hour, 12 hour, 24hour, 48 hour] |
| Wikipedia | 1 day | 2000 | 30 | 30 | [1 day, 4 day, 7 day, 14 day] |

Table 6: Detailed information of the datasets used in our benchmark including data frequency and number of times series (dimension), including the information about context length and prediction length and the multi-granularity dictionary utilized in the multivariate time series forecasting task.

## C.2 LIBRARIES USED

The **MG-TSD** code in this study is implemented using PyTorch (Paszke et al., 2019). It utilizes the PytorchTS library (Rasul, 2021), which enables convenient integration of PyTorch models with the GluonTS library (Alexandrov et al., 2020b) on which we heavily rely for data preprocessing, model training, and evaluation in our experiments.

The code for the baseline methods is obtained from the following sources.

(i) Vec-LSTM-ind-scaling: models the dynamics via an RNN and outputs the parameters of an independent Gaussian distribution with mean-scaling.
Code: `https://github.com/mbohlkeschneider/gluon-ts/tree/mv_release`;

(ii) GP-scaling: a model that unrolls an LSTM with scaling on each individual time series before reconstructing the joint distribution via a low-rank Gaussian.
Code: `https://github.com/mbohlkeschneider/gluon-ts/tree/mv_release`

(iii) GP-Copula: a model that unrolls an LSTM on each individual time series. The joint emission distribution is then represented by a low-rank plus diagonal covariance Gaussian copula.
Code: `https://github.com/mbohlkeschneider/gluon-ts/tree/mv_release`;

(iv) LSTM-MAF: a model which utilizes LSTM for modeling the temporal conditioning and employs Masked Autoregressive Flow (Papamakarios et al., 2017) for the distribution emission.
Code: `https://github.com/zalandoresearch/pytorch-ts/tree/master/pts/model/tempflow`

(v) Transformer-MAF: a model which utilizes Transformer (Vaswani et al., 2017) for modeling the temporal conditioning and employs Masked Autoregressive Flow (Papamakarios et al., 2017) for the distribution emission model.
Code: `https://github.com/zalandoresearch/pytorch-ts/tree/master/pts/model/transformer_tempflow`

(vi) TimeGrad: an auto-regressive model designed for multivariate probabilistic time series forecasting, assisted by an energy-based model.
Code: `https://github.com/zalandoresearch/pytorch-ts`

(vii) TACTiS: a non-parametric copula model based on transformer architecture.
Code: `https://github.com/servicenow/tactis`

(viii) D$^3$VAE: a bidirectional variational auto-encoder(BVAE) equipped with diffusion, denoise, and disentanglement.
Code: `https://github.com/ramber1836/d3vae`.

(ix) TimeDiff: a predictive framework trained by blending hidden contextual elements with future actual outcomes for sample conditioning.
Code: There is no publicly available code; we obtained the code by emailing the author.

(x) Autoformer: redefines the Transformer with a deep decomposition architecture, including sequence decomposition units, self-correlation mechanisms, and encoder-decoders.
Code: `https://github.com/thuml/Autoformer`

(xi) PatchTST: an efficient design of Transformer-based models for multivariate time series forecasting and self-supervised representation learning.
Code: `https://github.com/yuqinie98/PatchTST`

### C.3 HYPER-PARAMETER SETTING FOR EACH MODEL

| Dataset | Num gran | Gran dict | Share ratio | Loss weight |
|---|---|---|---|---|
| Solar Electricity Traffic KDD-cup | 2 | [1h,4h]
[1h,12h] | [1,0.9]
[1,0.8]
[1,0.6] | [0.9,0.1] |
| | 3 | [1h,4h,12h]
[1h,4h,24h] | [1,0.9,0.8]
[1,0.8,0.8]
[1,0.8,0.6] | [0.8, 0.1, 0.1]
[0.9, 0.05, 0.05]
[0.85, 0.10, 0.05] |
| | 4 | [1h,4h,12h,24h]
[1h,4h,12h,48h] | [1,0.9,0.8,0.8]
[1,0.9,0.8,0.6]
[1,0.8,0.6,0.6]
[1,0.8,0.6,0.4] | [0.8, 0.1, 0.05, 0.05]
[0.7,0.1,0.1,0.1] |
| | 5 | [1h,4h,8h,12h,24h]
[1h,4h,12h,24h,48h] | [1,0.9,0.8,0.6,0.6]
[1,0.9,0.8,0.6,0.4]
[1,0.8,0.6,0.6,0.6]
[1,0.8,0.6,0.6,0.4]
[1,0.8,0.6,0.4,0.4] | [0.8,0.1,0.05,0.04,0.01]
[0.8,0.05,0.05,0.05,0.05]
[0.6,0.1,0.1,0.1,0.1] |
| Taxi | 2 | [30m,2h]
[30m,6h]
[30m,12h]
[30m,24h] | [1,0.8]
[1,0.6] | [0.9,0.1] |
| Wikipedia | 3 | [1d,4d]
[1d,7d]
[1d,14d] | [1,0.8]
[1,0.6] | [0.9,0.1] |

Table 7: Tested hyper-parameter values for the **MG-TSD** Model. The reported results in the paper are based on a parameter search within these choices.

## D APPENDIX: METRICS

More details about the metrics we adopt can be found in Gluonts documentation (Alexandrov et al., 2020a). We briefly summarize them as below:

**CRPS$_{\text{sum}}$** : From de Bézenac et al. (2020), CRPS is a univariate strictly proper scoring rule which measures the compatibility of a cumulative distribution function $F$ with an observation $x \in \mathbb{R}$ as

$$\text{CRPS}(F, x) = \int_{\mathbb{R}} (F(y) - \mathbf{1}(x \le y))^2 \mathrm{d}y$$

where $\boldsymbol{I}\{x \le y\}$ is the indicator function, which is 1 if $x \le y$ and 0 otherwise. The CRPS attains the minimum value when the predictive distribution $F$ same as the data distribution. CRPS$_{\text{sum}}$ extends CRPS to multivariate time series with a simple modification.

$$\text{CRPS}_{\text{sum}} = \mathbb{E}_t[\text{CRPS}(F_{\text{sum}}^{-1}, \sum_i x_t^i)],$$

where $F_{\text{sum}}^{-1}$ is calculated by summing samples across dimensions and then sorted to get quantiles. A smaller $\text{CRPS}_{\text{sum}}$ indicates better performance.

**NMAE**: NMAE is a normalized version of the Mean Absolute Error (MAE) that takes into consideration the scale of the target values. The formula for NMAE is as follows:

$$\text{NMAE} = \frac{\text{mean}(|(\hat{Y} - Y)|)}{\text{mean}(|Y|)}$$

Similarly, in this formula, $\hat{Y}$ represents the predicted time series, and $Y$ represents the true target time series. NMAE calculates the average absolute difference between predictions and true values, normalized by the mean absolute magnitude of the target values. A smaller NMAE implies more accurate predictions.

**NRMSE**: NRMSE is a normalized adaptation of the Root Mean Squared Error (RMSE) that factors in the scale of the target values. The formula for NRMSE is as follows:

$$\text{NRMSE} = \sqrt{\frac{\text{mean}((\hat{Y} - Y)^2)}{\text{mean}(|Y|)}}$$

Here, $\hat{Y}$ represents the predicted time series, and $Y$ represents the true target time series. NRMSE measures the average squared difference between predictions and true values, normalized by the mean absolute magnitude of the target values. A smaller NRMSE indicates more accurate predictions.

# E  APPENDIX: MORE ILLUSTRATIVE PLOTS

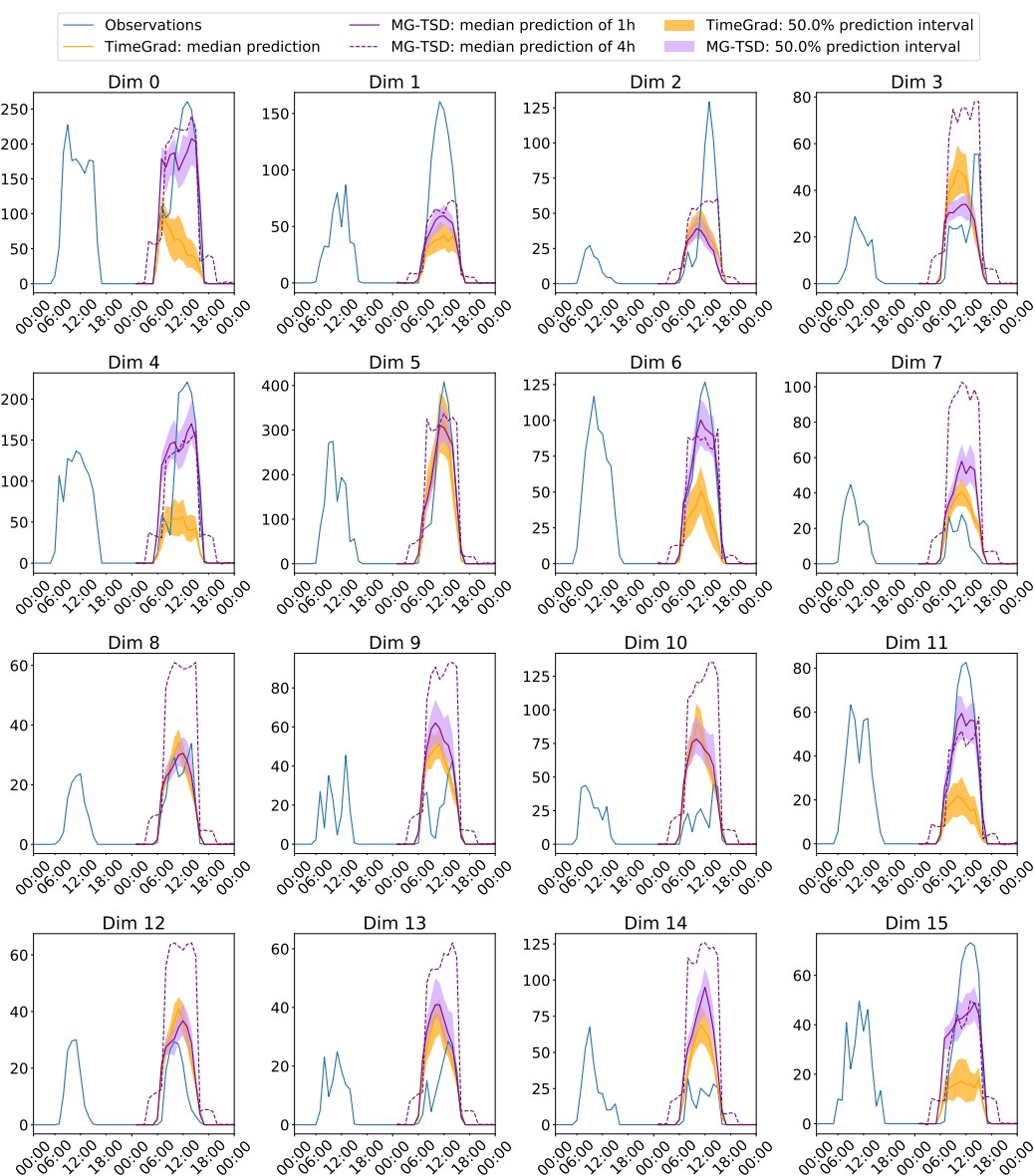

Figure 8: **MG-TSD** and TimeGrad prediction intervals and test set ground-truth for Solar data of some illustrative dimensions of 370 dimensions from first rolling-window.

