# OpenReview forum: "MG-TSD: Multi-Granularity Time Series Diffusion Models with Guided Learning Process"
_ICLR.cc/2024/Conference — ICLR 2024 poster_

### Official Review · Reviewer_FBCv · 2023-10-31

**Soundness:** 3 good
**Presentation:** 3 good
**Contribution:** 3 good
**Rating:** 6
**Confidence:** 4

**Summary:**

This work proposes a Multi-Granularity Time Series Diffusion (MG-TSD) model for time series prediction. In general, MG-TSD controls the learning process of the diffusion model by leveraging the temporal signals in time series data at different granularity levels. In particular, the authors link the forward process of the diffusion model to the data smoothing process. Motivated by this, this work develops a multi-granularity guidance diffusion loss function, such that the inherent features within data can be preserved and a regularized sampling path can be achieved. Experiments on real-world time series datasets demonstrate the effectiveness of the proposed approach.

**Strengths:**

1. In the context of time series forecasting, it is a good idea to stabilize the diffusion model with the help of coarse-grained temporal signals in time series data.
2. The derivation of the multi-granularity guidance loss function is solid. Besides, the learning procedure devised in this work is applicable.

**Weaknesses:**

1. The assumption that "... forward process of the diffusion model ... intuitively aligns with the process of smoothing fine-grained data into a coarser-grained representation ..." is not verified through theoretical analysis or empirical study.

2. Experimental settings for the time series forecasting task are unclear. For example, how the context interval and prediction interval are constructed in a time series dataset? Do you utilize a sliding window to roll the time series to build the context and predict intervals? Do the consecutive context/predict intervals overlap or not?

3. More experimental results are expected. For instance, the authors only evaluate the time series forecasting methods under one length setting in each dataset, like the setting of context-24-predict-24 is utilized in Solar, Electricity, Traffic, Taxi, and KDD-Cup datasets.

4. The reproductivity of this work is a concern.


\\======================\\

After rebuttal.

Most of my concerns are clarified.

**Questions:**

According to the inference procedure, the proposed time series forecasting approach can predict one future horizon time step at a time. How to apply this predictive approach for the long-term time series forecasting task effectively?

---

> ### Author Response · Authors · 2023-11-18
> **Response to Reviewer FBCv (1/3)**
>
> We sincerely appreciate your thorough review and valuable feedback. Enclosed are our responses to your insightful questions and concerns.
>
> **Q1. According to the inference procedure, the proposed time series forecasting approach can predict one future horizon time step at a time. How to apply this predictive approach for the long-term time series forecasting task effectively?**
>
> We are grateful to the reviewer for the insights.
>
> - In our Temporal Process Module, we utilize the RNN as the backbone model to encode the temporal dependence sequentially. Therefore, predicting m prediction length involves taking n-to-1 (n timesteps predicts one timestep) for m times autoregressively, where each inference step incorporates previous predictions into the context. Based on the extra experiment concerning long-term forecasting, our method experiences only a moderate decrease in performance while extending prediction length.
> - It is worth noting that our Temporal Process Module is adaptable and can accommodate other encoders capable of handling variable-length time steps. This includes, but is not limited to, autoregressive RNNs in an n-to-1 manner or non-autoregressive methods of an n-to-m manner such as transformers.
> - In certain scenarios, a Transformer-based TPM might be a more efficient solution, taking m timesteps predictions with one step. However, Transformers have limitations regarding the maximum length of the time series, which means the context length and prediction length cannot be increased infinitely. Additionally, for high-dimensional time series data (such as Wikipedia dataset with 2000 dimensions), the attention mechanism might suffer from huge training and inference overhead.
>
> Since the primary focus of this paper is on multi-granularity guidance, we did not test the eligibility of transformer variant for TPM and we will take it as an important future work.
>
> **W1. The assumption that "... forward process of the diffusion model ... intuitively aligns with the process of smoothing fine-grained data into a coarser-grained representation ..." is not verified through theoretical analysis or empirical study.**
>
> Thanks to the reviewer for the valuable insights. We would like to point out that the results depicted in Figure 3 of our manuscript provide empirical support for our assumption. Please note that, in Figure 3, the x-axis denotes the denoising diffusion steps, and the y-axis signifies the $\\text{CRPS}\_{\\text{sum}}$ values.
>
> Figure 3 illustrated an investigation into the effect of share ratio, where we evaluate the performance of MG-TSD using various share ratios across different coarse granularities.
>
> In Figure 3(a)-(d), the dashed blue curve in each plot represent $\\text{CRPS}\_{\\text{sum}}$ values between the coarse-grained targets and 1-hour samples come from 1-gran (finest-gran) model at each intermediate denoising step; each point on the orange polylines represents the $\\text{CRPS}\_{\\text{sum}}$ value of 1-hour predictions by 2-gran MG-TSD models (where one coarse granularity is utilized to guide the learning process for the finest-grained data), with different share ratios ranging from [0.2, 0.4, 0.6, 0.8, 1.0], and the lowest point of the line segment can be used to characterize the most suitable share ratio for the corresponding granularity.
>
> The four subplots from (a) to (d) illustrate a gradual smoothing transformation of the distribution of increasingly coarser targets. A key observation is that from the left to the right panel, the distribution of coarser targets gradually aligns with the distribution of intermediate samples at larger diffusion steps. More specifically, as granularity transitions from fine to coarse (4h→6h→12h→24h), the denoising steps at which the distribution most resembles the coarse-grained targets increase (approximately at steps 20→40→60→60). This comparison underscores the similarity between the diffusion process and the smoothing process from the finest-grained data to coarse-grained data, both of which involve a gradual loss of finer characteristics from the finest-grained data through a smooth transformation.

---

> > ### Comment · Reviewer_FBCv · 2023-11-22
> > **Question on response to W1**
> >
> > I am confused that, as the four subplots from (a) to (d) in Figure 3 suggested, the denoising diffusion steps at which the distribution most resembles the coarse-grained targets decrease, i.e., 80 $\rightarrow$ 60 $\rightarrow$ 40 $\rightarrow$ 40, which is opposite to your statement.
> > Please correct me if I missed any key points.

---

> > > ### Author Response · Authors · 2023-11-22
> > > **Response to "Question on response to W1"**
> > >
> > > Thank you very much for your response, and you are correct. There is indeed a typographical error in our previous message. In the statement "denoising steps at which the distribution most resembles the coarse-grained targets increase (approximately at steps 20→40→60→60)", the term "**denoising step**" should indeed be corrected to "**diffusion step**". What we intended to express was that the corresponding steps in the **diffusion process** are approximately 20→40→60→60, whereas in the **denoising process**, the corresponding steps are approximately 80→60→40→40, exactly as you pointed out. We sincerely apologize for this oversight in our response. Please be assured that our expressions in the manuscript are correct.

---

> > > > ### Comment · Reviewer_FBCv · 2023-11-22
> > > >
> > > > Thanks for your clarification.
> > > >
> > > > Again, I am still not well convinced that, regarding a time series, the connection between the diffusion process (forward process) and the smoothing process does hold. Could you provide more convincing evidence?
> > > >
> > > > From the point of view of time series decomposition, a time series $X$ can be approximated as $X = T + S + \epsilon$ where $T$ and $S$ denote trend and seasonality, respectively. And $\epsilon$ corresponds to the noisy part, which can also be deemed as details of $X$. As the diffusion steps increase, $X$ will be corrupted by $X^{\prime} = T + S + \epsilon + \epsilon^{\prime}$, how to prove that $X^{\prime}$ represents the coarser version of $X$?

---

> > > > > ### Author Response · Authors · 2023-11-23
> > > > > **Further explanation on the connection between forward diffusion process and smoothing process**
> > > > >
> > > > > Thanks for the quick response and your insights. We would like to highlight the similarity of the two processes lying in that the forward diffusion and smoothing process both lead to a loss of finer features of the original complex distribution of the fine-grained data.
> > > > >
> > > > > - In real-world applications, the finest-grained time series data exhibit significant fluctuations and complex temporal dynamics. According to additive decomposition $X=T+S+\\epsilon$, after extracting the trend and seasonality, the delicate fine features of the distribution are majorly encoded in the seasonality $S$ and residual $\\epsilon$ terms. During the smoothing process from fine-grained to coarse-grained, the seasonality term $S$ and the residual terms $\\epsilon$ distributions are susceptible to shifts and corruption, while the trend $T$ may be preserved when the window size is not too large. The superposition of diverse seasonality patterns and noise can cancel each other out, gradually losing the finer features in the previous decomposition.
> > > > >
> > > > > - As for the forward diffusion process, the decomposition can be rewritten as  $X^{\\prime}=\\sqrt{\\bar{\\alpha\_n}}(T+S+\\epsilon)+\\sqrt{1-\\bar{\\alpha\_n}}\\epsilon^{\\prime}=\\sqrt{\\bar{\\alpha\_n}}T+\\sqrt{\\bar{\\alpha\_n}}S+\\sqrt{\\bar{\\alpha\_n}}\\epsilon+\\sqrt{1-\\bar{\\alpha\_n}}\\epsilon^{\\prime}.$ The seasonality term $S$ can further be decomposed into the lower-frequency $S\_\\text{low}$ and the high-frequency $S\_\\text{high}$. The delicate fine features of the distribution are mainly encoded in the $S\_{\\text{high}}$. During the diffusion forward process, the $S\_{\\text{high}}$ term is gradually overwhelmed by larger white noise and is susceptible to corruption in the first few diffusion steps. The lower-frequency seasonality and the trend term are preserved within a certain number of diffusion steps.
> > > > >
> > > > > - For a better illustration, we sampled series from Solar dateset and we conducted a Fast Fourier Transform to extract the seasonality components of the series, as well as the samples of different granularities and corresponding noisy samples along the forward diffusion process. Results are appended in the **appendix B.2.3** of our manuscript. As depicted in Figure 7(a), as granularity becomes coarser, the components of all outstanding frequencies get lower, while the high-frequency peak (around 125 and 80) diminishes quicker than lower-frequency peak (around 45). Figure 7(b) demonstrates the distribution of frequency components of the same noisy series with gradually ascending forward diffusion steps and the same pattern is observable. This result further indicates the connection of the two processes in losing finer informative features, which motivated us to utilize coarse granularity data as guidance and the main experiment in our manuscript demonstrates performance gain brought by incorporating multi-granularity guidance.

---

> ### Author Response · Authors · 2023-11-18
> **Response to Reviewer FBCv (2/3)**
>
> **W2. Experimental settings for the time series forecasting task are unclear.**
>
> Thanks the reviewer for the question. To clarify, we generate coarse-grained data from the finest-grained time series data over the entire timeline. We rely on the GluonTS[1] library for data splitting and creating the training and testing instances. The GluonTS library is widely used in time series forecasting. Further explanations are provided below:
>
> - Generation of the coarse-grained data: Section 3.1 has the details to generate multi-granularity data. A sliding window with a pre-defined size $s^g$ for granularity $g$ is applied to the finest-grained data across the entire timeline, where $g=1,2,...,G$. These **sliding windows** are intentionally **non-overlapping**. Within each window, we smooth the finest-grained data by averaging and replicate the average $s^g$ times to align with the timeline $[1, T]$.
> - Dataset splitting: All the datasets we used in the benchmark experiments are public and available in GluonTS[1]. The library has these datasets pre-split into training and testing sets, and we follow this splitting method to obtain our training and testing datasets.
> - Creation of training and testing instances: we randomly sample the context window, followed by the prediction window, from the complete training data. This process can be viewed as applying a moving window to auto-regressively roll through the entire timeline, with consecutive time intervals **overlapping.** Furthermore, for different datasets, the context length and prediction length vary, such as 24-hour-24-hour for Electricity, and 30-Day-30-Day for Wikipedia, as detailed in Appendix C.1.
>
> **W3. More experimental results about different prediction lengths are expected.**
>
> Thanks the reviewer for insightful suggestion. We have conducted more experiments to test the performance of MG-TSD method with different prediction length.
>
> **Experiment setting**: In the current experiment, we assess the long-term prediction performance of MG-TSD by comparing it with the baseline TimeGrad, chosen for its competitive performance in the main experiment (Table 1) and its shared characteristic of forecasting in an autoregressive manner.  For the solar and electricity datasets (with the frequency of 1 hour), we set the context length to 24 hours and evaluate the performance of methods with prediction lengths of 24 hours, 48 hours, 96 hours and 144 hours. The average $\\text{CRPS}\_{\\text{sum}}$, $\\text{NRMSE}\_{\\text{sum}}$, and $\\text{NMAE}\_{\\text{sum}}$ metrics are computed for both MG-TSD and the baseline over 10 independent runs, with error bars indicating the corresponding standard deviations. (We would include additional baselines in the plots later to strengthen our conclusions for updated paper. )
>
> **Experiment results and findings**: The preliminary results indicate that MG-TSD performs well for long-time forecasting. The results indicate that as the prediction length increases, the performance of our proposed method stays robust, exhibiting no sudden decline. Furthermore, our method consistently outperforms the competitive baseline. This performance advantage is anticipated to persist in future trends, with no indication of convergence between the approaches.
>
> **Table G: Results for Solar Dataset**
>
> |Prediction Length|Method|$\\text{CRPS}\_{\\text{sum}}$|$\\text{NRMSE}\_{\\text{sum}}$|$\\text{NMAE}\_{\\text{sum}}$|
> |-|-|-|-|-|
> |24h|TimeGrad|0.3335±0.0653|0.6952±0.0644|0.3637±0.0665|
> |48h|TimeGrad|0.3615±0.0298|0.7392±0.0566|0.4070±0.0298|
> |96h|TimeGrad|0.3737±0.0213|0.7905±0.0481|0.4113±0.0238|
> |144h|TimeGrad|0.4301±0.0140|0.9285±0.0219|0.4768±0.0109|
> |24h|MG-TSD|0.3178±0.0342|0.6591±0.0503|0.3480±0.0356|
> |48h|MG-TSD|0.3401±0.0271|0.7234±0.0398|0.3862±0.0217|
> |96h|MG-TSD|0.3500±0.0270|0.7395±0.0439|0.3909±0.0264|
> |144h|MG-TSD|0.3659±0.0311|0.8179±0.0600|0.4226±0.0292|
>
> **Table H: Results for Electricity Dataset**
>
> |Prediction Length|Method|$\\text{CRPS}\_{\\text{sum}}$|$\\text{NRMSE}\_{\\text{sum}}$|$\\text{NMAE}\_{\\text{sum}}$|
> |-|-|-|-|-|
> |24h|TimeGrad|0.0205±0.0033|0.0348±0.0057|0.0266±0.0049|
> |48h|TimeGrad|0.0264±0.0020|0.0474±0.0034|0.0343±0.0026|
> |96h|TimeGrad|0.0304±0.0048|0.0558±0.0092|0.0407±0.0065|
> |144h|TimeGrad|0.0532±0.0090|0.0953±0.0153|0.0674±0.0096|
> |24h|MG-TSD|0.0174±0.0042|0.0296±0.0086|0.0226±0.0071|
> |48h|MG-TSD|0.0212±0.0028|0.0334±0.0045|0.0279±0.0042|
> |96h|MG-TSD|0.0224±0.0069|0.0376±0.0103|0.0286±0.0086|
> |144h|MG-TSD|0.0341±0.0091|0.0609±0.0142|0.0473±0.0116|
>
> **Action taken:** We have included the experiments and results in Appendix B.2. Figure 5 in the appendix visualizes the results in Table G and Table H provided here, providing a clearer presentation of our conclusions.

---

> ### Author Response · Authors · 2023-11-18
> **Response to Reviewer FBCv (3/3)**
>
> **W4. Reproducibility**
>
> We plan to release our code soon. We are intensively working on cleaning up our code to ensure it is well-documented, thoroughly tested, and user-friendly. Additionally, to guarantee the reproducibility of our experiments, we performed 10 independent runs for each setting and reported both the average values and the standard deviation (std).
>
> References:
>
> [1]Alexandrov A, Benidis K, Bohlke-Schneider M, et al. Gluonts: Probabilistic and neural time series modeling in python[J]. The Journal of Machine Learning Research, 2020, 21(1): 4629-4634.

---

### Official Review · Reviewer_Zqn4 · 2023-11-01

**Soundness:** 3 good
**Presentation:** 4 excellent
**Contribution:** 3 good
**Rating:** 6
**Confidence:** 4

**Summary:**

Diffusion probabilistic models which can generate high-fidelity samples reserve stochastic nature. This characteristic makes it less effective in probabilistic time series forecasting tasks. To improve the efficiency of Diffusion probabilistic models, this paper introduces a novel MG-TSD model with an innovatively designed multi-granularity guidance loss function that efficiently guides the diffusion learning process. To effectively utilize coarse-grained data across various granularity levels, this paper propose a concise implementation method. What’s more, this approach does not rely on additional external data, making it versatile and applicable across various domains. Extensive experiments conducted on real-world datasets demonstrate the superiority of the proposed model, achieving the best performance compared to the state-of-the-art methods.

**Strengths:**

1. In the context of the time series forecasting, where fixed observations exclusively serve as objectives, diffusion probabilistic models would result in forecasting instability and inferior prediction performance. Unlike constraining the intermediate states during the sampling process, this paper creatively leverages multiple granularity levels within data to guide the learning process of diffusion models.
2. This paper provides a series of ablation experiments to test the effect of share ratio and the number of granularities, it evaluate the performance of MG-TSD using various share ratios across different coarse granularities and the number of granularities.
3. Clarity:  The paper offers a clear presentation to the model architecture with a good explanation of the methodology.

**Weaknesses:**

1. This paper lacks an evaluation of the time complexity of the model. It may be more sufficient to add experiments that consume memory and time.
2. MG-TSD is consisting of Multi-granularity Data Generator, Temporal Process Module (TPM), and Guided Diffusion Process Module. However, the ablation experiment part of this paper lacks performance testing of each module, especially Multi-granularity

**Questions:**

1. In Equation 7 of Section 3.2.1, is the distribution of ∈ consistent with the distribution of x_N? Does the distribution of ∈ obey the normal distribution? The distribution and meaning of ∈ are not pointed out.
2. Compared with the existing diffusion probabilistic models, does the MG-TSD model framework differs only in the Guided Diffusion Process Module ?

---

> ### Author Response · Authors · 2023-11-18
> **Response to Reviewer Zqn4 (1/2)**
>
> We are grateful for your constructive comments. Below we address each question and concern.
>
> **Q1: In Equation 7 of Section 3.2.1, is the distribution of $\\epsilon$ consistent with the distribution of $x\_N$? Does the distribution of $\\epsilon$ obey the normal distribution? The distribution and meaning of $\\epsilon$ are not pointed out.**
>
> Thank the reviewer for bringing attention to this. The $\\epsilon$ in equation (7) adheres to the standard normal distribution, consistent with the backbone denoising diffusion models. The dimension of the  $\\epsilon$ is the same with $x^{g}$.
>
> **Action taken**: We have incorporated this clarification into our manuscript.
>
> **Q2: Compared with the existing diffusion probabilistic models, does the MG-TSD model framework differs only in the Guided Diffusion Process Module?**
>
> Thanks for this question. We would like to further clarify on our module design.
>
> - The core difference between MG-TSD and previous work principally lies in the approach of incorporating information from data. We creatively propose a multi-granularity guidance approach to naturally exploit intrinsic coarse-to-grain features within data to stabilize the samples from the diffusion model. This is facilitated by the Guided Diffusion Process Module.
> - The multi-granularity data generator is unique in our work for the preprocessing of original data into multiple alternatives with different granularities. This module collaborates synergistically with the Guided Diffusion Process Module, enabling it to adapt to multi-granularity requirements.
> - Although the temporal process module is similar to previous work[1][2] in its function for extracting historical context, it was dedicatedly designed to adapt for the multi-granularity cases. To be more specific, temporal process module employs separate RNN submodule for each granularity, without parameters sharing across different granularities.
>
> **W1. Concern about the time complexity and memory of the model.**
>
> Experiments have been conducted to evaluate the time and memory usage of the MG-TSD model during training across various granularities.
>
> **Experiment setting:** These experiments were executed using a single A6000 card with 48G memory capacity. The Solar dataset was utilized in this context, with a batch size of 128, input size of 552, 100 diffusion steps, and 30 epochs.
>
> **Experiment results and findings:** As illustrated in the corresponding graph and table, there is a linear increase in memory consumption with an increase in granularity. A slight addition in training time is also observed. These findings are coherent with the architecture of our model. In particular, each additional granularity results in the introduction of an extra RNN in the Temporal Process Module and an increase in computation within the Guided Diffusion Process Module. As per theoretical expectations, these resource consumptions should exhibit linear growth. Moreover, it is pertinent to mention that an excessive increase in granularity may not notably boost the final prediction results, hence the granularity should be kept within a certain range. Therefore, the consumption of memory will not rise indefinitely.
>
> **Table F: Comparison of Time and Memory Consumption at Different Granularity Levels in MG-TSD Model Training**
>
> |Granularity|Memory(GB)|RunTime(Minute)|Relative Memory Occupancy|Relative Run Time|
> |-|-|-|-|-|
> |2|6.06|25.93|100%|100%|
> |3|13.45|27.48|222%|106%|
> |4|20.03|33.33|331%|129%|
> |5|29.09|33.52|480%|129%|
>
>
> **Summary:** In summary, while it is evident that an increase in granularity escalates the consumption of both time and memory, these increases are reasonable and fall within acceptable boundaries.
>
> **Action taken:** We have included the experiments and results in Appendix B.2. Figure 6 in the appendix visualizes the results in Table F provided here, providing a clearer presentation of our conclusions.

---

> ### Author Response · Authors · 2023-11-18
> **Response to Reviewer Zqn4 (2/2)**
>
> **W2. Concern about the ablation study for the Multi-granularity Data Generator module and TPM module**
>
> Thanks to the reviewer for the comment.
>
> We would like to highlight that Table 3 in our manuscript serves as an ablation study of the Multi-granularity component in the MG-TSD model. The experiment, presented in Table 3, explores the effect of varying the number of granularity levels on the model's performance. Testing the performance of the model with an additional granularity level inherently assesses the effectiveness of all three modules. More specifically, the Multi-granularity Data Generator Module is executed to generate the data at a new granularity level. The Temporal Process Module (TPM) incorporates an additional RNN cell to encode the temporal pattern within the added granularity data, while the Guided Diffusion Process Module adapts the learning and generation process with the additional coarse-grained data. The results in Table 3 reveal that increasing the number of granularity levels typically improves the performance of the MG-TSD model. This finding confirms that the mechanism currently designed to generate coarse-grained data is effective in enhancing performance. Theoretically, the MG-TSD model can handle an unlimited increase in granularity levels. However, in practical scenarios, we observe that the marginal benefit tends to diminish as the number of granularity levels increases. Our findings suggest that employing four to five granularity levels typically suffices to achieve optimal performance.
>
> To further clarify, the Multi-granularity Generator Module functions as a predefined data pre-processing step and is utilized to generate coarse-grained data at various granularity levels. It does not involve parameter optimization during the training stage. The Temporal Process Module serves as an encoder in the model, which captures and compresses the temporal dependencies in time series data up to a certain timestep. The RNN is adopted in our model as it presents a feasible and convenient method of implementation for this module.  The choice of RNN as the backbone encoder in TPM aligns with various previous works, including TimeGrad[1] and GP-copula[2]. These studies utilize RNN for modeling temporal dependencies, which attests to the effectiveness of RNN. Since the primary focus of this paper is on multi-granularity guidance, we did not conduct extra ablation studies on the designs of these components.
>
> References:
>
> [1] Rasul K, Seward C, Schuster I, et al. Autoregressive denoising diffusion models for multivariate probabilistic time series forecasting[C]//International Conference on Machine Learning. PMLR, 2021: 8857-8868.
>
> [2] Salinas D, Bohlke-Schneider M, Callot L, et al. High-dimensional multivariate forecasting with low-rank Gaussian copula processes[J]. Advances in neural information processing systems, 2019, 32.

---

### Official Review · Reviewer_Zq3E · 2023-11-02

**Soundness:** 2 fair
**Presentation:** 3 good
**Contribution:** 2 fair
**Rating:** 6
**Confidence:** 4

**Summary:**

This paper employs the diffusion model for time series forecasting and introduces the Multi-Granularity Time Series Diffusion model, which comprises three key components: 1). The Multi-Granularity Data Generator, responsible for generating multi-granularity data. 2). The Temporal Process Module, which utilizes an RNN architecture to capture temporal dynamics. 3). The Guided Diffusion Process Module is aimed at generating stable time-series predictions. This model leverages various levels of granularity within the data to guide the forward process of the diffusion model. Additionally, the paper designs a multi-granularity guidance loss function and explores optimal configurations for different granularity levels, proposing a practical rule of thumb. Extensive experiments are conducted to showcase its precision and effectiveness.

**Strengths:**

1. The research problem addressed in this study is of paramount significance and holds great interest. Accurate time prediction has broad applications, including tasks like anomaly detection and energy consumption control.
2. The paper introduces a novel and intriguing approach by linking various granularities in the time series with the forward process in the diffusion model.
3. The paper is excellently written and presented in a clear and comprehensible manner.

**Weaknesses:**

1. Some related works can be further discussed.
2. There is only one metric in the main experiment, which is not enough.
3. Compared with the baseline, the performance improvement is not obvious.
4. The use of the RNN architecture requires further explanation

## After rebuttal
most of my concerns have been addressed.

**Questions:**

1.	There have been some similar works, such as TimeDiff[1] and D3VAE[2], which also applies the diffusion model. What are the technical advantages of these studies?
2.	The paper designed MG-TSD based on the diffusion model. Why is it not compared with the diffusion-based models in the baseline? Besides, There are some newer Transformer-based models, such as PatchTST[3], and Autoformer[4] should be compared in your experiments.
- [1]Shen L, Kwok J. Non-autoregressive Conditional Diffusion Models for Time Series Prediction[J]. arXiv preprint arXiv:2306.05043, 2023.
- [2] Li Y, Lu X, Wang Y, et al. Generative time series forecasting with diffusion, denoise, and disentanglement[J]. Advances in Neural Information Processing Systems, 2022, 35: 23009-23022.
- [3] Nie Y, Nguyen N H, Sinthong P, et al. A Time Series is Worth 64 Words: Long-term Forecasting with Transformers[C]//The Eleventh International Conference on Learning Representations. 2022.
- [4] Wu H, Xu J, Wang J, et al. Autoformer: Decomposition transformers with auto-correlation for long-term series forecasting[J]. Advances in Neural Information Processing Systems, 2021, 34: 22419-22430.

---

> ### Author Response · Authors · 2023-11-18
> **Response to Reviewer Zq3E (1/3)**
>
> Thank you very much for your careful review and constructive suggestions! Please find our response to your questions and concerns.
>
> **Q1&W1. Some related works can be further discussed. There have been some similar works, such as TimeDiff[1] and D3VAE[2], which also apply the diffusion model. What are the technical advantages of these studies?**
>
> Diffusion-based methods that use generations from conditional distribution as predictions are typically from probabilistic methods. In contrast, sequence-based (including transformer-based) models are mostly deterministic methods.
>
> Autoformer and PatchTST are deterministic time-series forecasting models based on dedicated designed Transformers, which are different from MG-TSD in the scope of prediction. D3VAE and TimeDiff are probabilistic time-series forecasting methods, which are closer to the scope of our work.
>
> Both D3VAE and TimeDiff involve diffusion probabilistic models. D3VAE utilizes only the forward diffusion process and the prediction stage is majorly taken over by a VAE architecture, which is fundamentally different from MG-TSD in the way of generating forecasts.
>
> In contrast, both TimeDiff and MG-TSD utilize a conditional denoising process over time intervals to perform forecasting. The major difference between TimeDiff and MG-TSD lies in the way of incorporating information within data, TimeDiff is trained with the mixed up of hidden contexts and future ground truths as sample conditionings, while MG-TSD utilizes multi-granularity guidance, leveraging intrinsic coarse-to-grain features within data.
>
> The technical advantages of diffusion-based time series forecasting are summarized as below:
>
> - **Uncertainty quantification.** A primary advantage of probabilistic models is their capability to accurately capture data distribution instead of just point estimates. This allows for the convenient construction of prediction intervals using multiple outputs from the diffusion model. By knowing the data distribution at a specific timestamp, one can quantify the prediction uncertainty and provide more reliable forecasts. It also enables the convenient evaluation of the probability of extreme events occurring. For instance, in the wind power sector, reliable forecasts are essential. An unexpected extreme event causing a wind farm to shut down can wipe out months of revenue. This underscores the importance of probabilistic forecasting, which considers both the expected power output and the uncertainty of the forecast, thereby aiding in the minimization of such risks [1].
>
> - **The capability to model arbitrary distributions without parametric assumptions.** This capacity of diffusion models to characterize distributions is also applicable to modeling the distribution of time series. A flaw in other methods of distribution modeling is that they are strictly constrained by the functional structure of their target distributions. For example, the previous choice, Transformer-MAF[2], models multivariate time series with an autoregressive deep learning model, in which the data distribution is expressed by a conditional normalizing flow. Diffusion-based methods, conversely, can offer a less restrictive solution, as indicated in reference [3].
>
> **Action Taken**: We have included additional experimental results with respect to these mentioned works, on datasets of different dimensions and predicting lengths. Overall, the performance of MG-TSD is superior. Please refer to the reply to **Q2&W2** below for numerical results. We have also included these results in the updated manuscript.
>
> **Q2&W2. The paper designed MG-TSD based on the diffusion model. Why is it not compared with the diffusion-based models in the baseline? Besides, there are some newer Transformer-based models, such as PatchTST[6], and Autoformer[7] should be compared in your experiments. There is only one metric in the main experiment, which is not enough.**
>
> Thanks for suggesting additional references. In our original benchmark experiment, the TimeGrad method included in our baseline is a diffusion-based model. To ensure comprehensive comparisons, we have now incorporated all the mentioned works, TimeDiff[4], D3VAE[5],  PatchTST[6], and AutoForemer[7], into our baselines.
>
> Furthermore, we have broadened our evaluation metrics to include $\\text{NMAE}\_{\\text{sum}}$ (Normalized MAE) and $\\text{NRMSE}\_{\\text{sum}}$ (Normalized RMSE). Detailed results for these additional metrics can now be found in the updated appendix, particularly in Tables 4 and 5.
>
> Here, we attach the benchmark experiment results of three metrics $\\text{CRPS}\_{\\text{sum}}$, $\\text{NMAE}\_{\\text{sum}}$, and $\\text{NRMSE}\_{\\text{sum}}$ below. These additional experiments validate that our method consistently delivers state-of-the-art performance.

---

> ### Author Response · Authors · 2023-11-18
> **Response to Reviewer Zq3E (2/3)**
>
> **Table A: Comparison of $\\text{CRPS}\_{\\text{sum}}$ (smaller is better) of models on six real-world datasets. The
> reported mean and standard error are obtained from 10 re-training and evaluation independent runs.**
> |Method|Solar|Electricity|Traffic|KDD-cup|Taxi|Wikipedia|Avg Rank|
> |-|-|-|-|-|-|-|-|
> |Vec-LSTM ind-scaling|0.4825±0.0027|0.0949±0.0175|0.0915±0.0197|0.3560±0.1667|0.4794±0.0343|0.1254±0.0174|8.8|
> |GP-Scaling|0.3802±0.0052|0.0499±0.0031|0.0753±0.0152|0.2983±0.0448|0.2265±0.0210|0.1351±0.0612|7.3|
> |GP-Copula|0.3612±0.0035|0.0287±0.0005|0.0618±0.0018|0.3157±0.0462|0.1894±0.0087|0.0669±0.0009|5.7|
> |LSTM-MAF|0.3427±0.0082|0.0312±0.0046|0.0526±0.0021|0.2919±0.1486|0.2295±0.0082|0.0763±0.0051|5.3|
> |Transformer-MAF|0.3532±0.0053|0.0272±0.0017|0.0499±0.0011|0.2951±0.0504|0.1531±0.0038|0.0644±0.0037|4.0|
> |TimeGrad|0.3335±0.0653|0.0232±0.0035|0.0414±0.0112|0.2902±0.2178|0.1255±0.0207|0.0555±0.0088|2.2|
> |D3VAE|0.4449±0.0375|0.1424±0.0883|0.3967±0.1165|0.4861±0.0517|0.5909±0.4645|1.9950±1.9874|10.0|
> |TimeDiff|1.3323±0.0305|0.3505±0.0075|0.4778±0.0058|0.3622±0.0127|0.4517±0.0101|0.1140±0.0105|9.7|
> |TACTiS|0.4209±0.0330|0.0259±0.0019|0.1093±0.0076|0.5406±0.1584|0.2070±0.0159|-|8.2|
> |MG-Input|0.3239±0.0427|0.0238±0.0035|0.0658±0.0065|0.2977±0.1163|0.1592±0.0087|0.0567±0.0091|3.8|
> |MG-TSD|**0.3081±0.0099**|**0.0149±0.0017**|**0.0323±0.0125**|**0.1837±0.0865**|**0.1159±0.0132**|**0.0529±0.0054**|**1.0**|
>
> **Table B: Comparison of $\\text{NRMSE}\_{\\text{sum}}$ (smaller is better) of models on six real-world datasets. The
> reported mean and standard error are obtained from 10 re-training and evaluation independent runs.**
> |Method|Solar|Electricity|Traffic|KDD-cup|Taxi|Wikipedia|Avg Rank|
> |-|-|-|-|-|-|-|-|
> |Vec-LSTM ind-scaling|0.9952±0.0077|0.1439±0.0228|0.1451±0.0248|0.4461±0.1833|0.6398±0.0390|0.1618±0.0162|7.2|
> |GP-Scaling|0.9004±0.0095|0.0811±0.0062|0.1469±0.0181|0.3445±0.0621|0.3598±0.0285|0.1710±0.1006|6.3|
> |GP-Copula|0.8279±0.0053|0.0512±0.0009|0.1282±0.0033|**0.2605±0.0227**|0.3125±0.0113|0.0930±0.0076|4.0|
> |Autoformer|0.7046±0.0000|0.0475±0.0000|0.0951±0.0000|0.8984±0.0000|0.3498±0.0000|0.1052±0.0000|5.2|
> |PatchTST|0.7270±0.0000|0.0474±0.0000|0.1897±0.0000|0.5137±0.0000|0.3690±0.0000|0.0915±0.0000|5.3|
> |D3VAE|0.7472±0.0508|0.1640±0.0928|0.4722±0.1197|0.5628±0.0419|0.7624±0.5598|2.2094±2.1646|8.3|
> |TimeDiff|1.5985±0.0359|0.3714±0.0073|0.5520±0.0087|0.4955±0.0147|0.5479±0.0084|0.1412±0.0099|8.3|
> |TimeGrad|0.6953±0.0845|0.0348±0.0057|0.0653±0.0244|0.4092±0.1332|0.2365±0.0386|0.0870±0.0106|2.3|
> |TACTiS|0.8532±0.0851|0.0427±0.0023|0.2270±0.0159|0.6513±0.1767|0.3387±0.0097|-|6.8|
> |MG-TSD|**0.6178±0.0418**|**0.0241±0.0030**|**0.0563±0.0230**|0.3001±0.0997|**0.2334±0.0313**|**0.0810±0.0057**|**1.2**|
>
> **Table C: Comparison of $\\text{NMAE}\_{\\text{sum}}$ (smaller is better) of models on six real-world datasets. The reported mean and standard error are obtained from 10 re-training and evaluation independent runs.**
>
> |Method|Solar|Electricity|Traffic|KDD-cup|Taxi|Wikipedia|Avg Rank|
> |-|-|-|-|-|-|-|-|
> |Vec-LSTM ind-scaling|0.5091±0.0027|0.1261±0.0211|0.1042±0.0228|0.4193±0.1902|0.4974±0.0351|0.1416±0.0180|7.3|
> |GP-Scaling|0.4945±0.0065|0.0648±0.0046|0.0975±0.0163|0.2892±0.0550|0.2867±0.0264|0.1452±0.1029|6.0|
> |GP-Copula|0.4302±0.0046|0.0312±0.0007|0.0769±0.0022|**0.2140±0.0124**|0.2390±0.0098|0.0659±0.0061|3.3|
> |Autoformer|0.6368±0.0000|0.0388±0.0000|0.0684±0.0000|0.7658±0.0000|0.2652±0.0000|0.1239±0.0000|6.5|
> |PatchTST|0.4351±0.0000|0.0350±0.0000|0.1219±0.0000|0.4497±0.0000|0.2887±0.0000|0.0625±0.0000|5.3|
> |D3VAE|0.4457±0.0377|0.1434±0.0892|0.3992±0.1177|0.4874±0.0520|0.6080±0.5061|2.0151±2.0005|8.5|
> |TimeDiff|1.3343±0.0305|0.3519±0.0075|0.4782±0.0058|0.3630±0.0127|0.4521±0.0102|0.1146±0.0106|8.0|
> |TimeGrad|0.3694±0.0400|0.0266±0.0049|0.0410±0.0089|0.3614±0.1334|0.1365±0.0193|0.0631±0.0080|2.5|
> |TACTiS|0.4448±0.0313|0.0310±0.0015|0.1352±0.0159|0.6078±0.1718|0.2244±0.0036|-|6.3|
> |MG-TSD|**0.3347±0.0220**|**0.0178±0.0018**|**0.0370±0.0140**|0.2463±0.0865|**0.1300±0.0150**|**0.0601±0.0057**|**1.2**|
>
> **W3. Concerns about the significance of the model’s improvements compared to baselines.**
>
> For clarity, the original Table 1 results illustrate the MG-TSD model's performance with two granularities. Our results in Table 3 suggests employing more granularity levels could lead to further performance improvements. We have updated results in Tables 1, 4 and 5 to better reflect the MG-TSD model's optimal performance using multiple granularities.

---

> ### Author Response · Authors · 2023-11-18
> **Response to Reviewer Zq3E (3/3)**
>
> We provide more statistics summaries to evaluate the performance improvement.  We have recorded the rankings of different methods based on their performance on various datasets. The results are shown in table below.
>
> **Table D: Performance ranking of methods across datasets**
> |Method|$\\textbf{CRPS}\_{\\textbf{sum}}$ Avg Rank|$\\textbf{NRMSE}\_{\\textbf{sum}}$ Avg Rank|$\\textbf{NMAE}\_{\\textbf{sum}}$ Avg Rank|
> |-|-|-|-|
> |Vec-LSTM ind-scaling|8.8|7.2|7.3|
> |GP-Scaling|7.3|6.3|6.0|
> |GP-Copula|5.7|4.0|3.3|
> |Autoformer|5.3|5.2|6.5|
> |PatchTST|4.0|5.3|5.3|
> |D3VAE|2.2|8.3|8.5|
> |TimeDiff|10.0|8.3|8.0|
> |TimeGrad|9.7|2.3|2.5|
> |TACTiS|8.2|6.8|6.3|
> |MG-TSD|1.0|1.2|1.2|
>
> From the results, our method achieved the best rank among all baselines on six datasets in terms of the $\\text{CRPS}\_{\\text{sum}}$.  As for the $\\text{NRMSE}\_{\\text{sum}}$ and $\\text{NMAE}\_{\\text{sum}}$ metrics, we achieved the best performance on all datasets, except for KDD-cup dataset where the GP-Copular method was slightly better.
>
> Additionally, we have recorded the percentage improvement of the MG-TSD method relative to the top-performing baseline in the table below for $\\text{CRPS}\_{\\text{sum}}$.
>
> **Table E: Relative improvement of MG-TSD method over top-performing baseline**
> |Metric|Solar|Electricity|Traffic|KDD-cup|Taxi|Wikipedia|
> |-|-|-|-|-|-|-|
> |$\\text{CRPS}\_{\\text{sum}}$|7.6%|35.8%|22.0%|36.7%|7.6%|4.7%|
>
> On the six datasets, we achieved $\\text{CRPS}\_{\\text{sum}}$ improvements, ranging from 4.7% to 35.8%. The modest improvements were achieved on the Wikipedia dataset. This might be due to its high-dimensional nature. The improvement of our model is not trivial, given the challenges posed by these datasets.
>
> **Action Taken**: In Tables 1, 4, and 5, we have updated the results to reflect the best performance of the MG-TSD model with multiple granularities, rather than just two.
>
> **W4. The use of the RNN architecture requires further explanation.**
>
> Thanks for the question. To clarify, the Temporal Process Module serves as an encoder in the model, which captures and compresses the temporal dependencies in time series data up to a certain timestep. The Recurrent Neural Network (RNN) is adopted in our model as it presents a feasible and convenient method of implementation for this module. It is worth noting that our Temporal Process Module is adaptable and can accommodate other encoders capable of handling variable-length time steps. This includes, but is not limited to, autoregressive RNNs in an n-to-1 manner or transformers in an n-to-m manner.
>
> Furthermore, the choice of RNN as the backbone encoder in TPM aligns with various previous works, including TimeGrad[8] and GP-copula[9]. These studies utilize RNN for modeling temporal dependencies, which attests to the effectiveness of RNN. Notably, differing from these existing works, which typically use a single RNN for single-granularity data, we employ multiple RNNs in the model to leverage the pattern and temporal information in time series data at various granularity levels. These RNNs operate without parameter sharing and are trained simultaneously with the Guided Diffusion Process Module.
>
>
> References:
>
> [1]Bjerregård M B, Møller J K, Madsen H. An introduction to multivariate probabilistic forecast evaluation[J]. Energy and AI, 2021, 4: 100058.
>
> [2]Rasul K, Sheikh A S, Schuster I, et al. Multivariate probabilistic time series forecasting via conditioned normalizing flows[J]. arXiv preprint arXiv:2002.06103, 2020.
>
> [3]Lin L, Li Z, Li R, et al. Diffusion models for time series applications: A survey[J]. arXiv preprint arXiv:2305.00624, 2023.
>
> [4]Shen L, Kwok J. Non-autoregressive Conditional Diffusion Models for Time Series Prediction[J]. arXiv preprint arXiv:2306.05043, 2023.
>
> [5] Li Y, Lu X, Wang Y, et al. Generative time series forecasting with diffusion, denoise, and disentanglement[J]. Advances in Neural Information Processing Systems, 2022, 35: 23009-23022.
>
> [6] Nie Y, Nguyen N H, Sinthong P, et al. A Time Series is Worth 64 Words: Long-term Forecasting with Transformers[C]//The Eleventh International Conference on Learning Representations. 2022.
>
> [7] Wu H, Xu J, Wang J, et al. Autoformer: Decomposition transformers with auto-correlation for long-term series forecasting[J]. Advances in Neural Information Processing Systems, 2021, 34: 22419-22430.
>
> [8] Rasul K, Seward C, Schuster I, et al. Autoregressive denoising diffusion models for multivariate probabilistic time series forecasting[C]//International Conference on Machine Learning. PMLR, 2021: 8857-8868.
>
> [9] Salinas D, Bohlke-Schneider M, Callot L, et al. High-dimensional multivariate forecasting with low-rank gaussian copula processes[J]. Advances in neural information processing systems, 2019, 32.

---

### Author Response · Authors · 2023-11-18
**Response to all reviewers**

We thank all the reviewers for the effort engaged in the review phase. We really appreciate those constructive comments and insights!  We are truly grateful for the reviewers' positive feedback on the innovative nature of our methods, our presentation and derivations, and our detailed ablation study.

Based on these valuable comments, we have made the following revisions in our updated manuscript.

- We have conducted additional benchmark experiments. The new results are summarized and now included in the revised supplementary material, Appendix B.1 (Tables 4 and 5). Below is a brief summary of what we have done:
- We have added four new baselines: TimeDiff, D3VAE, PatchTST, and Autoformer.
- We have included two additional evaluation metrics to the main experiment: $\\text{NRMSE}\_{\\text{sum}}$ and $\\text{NMAE}\_{\\text{sum}}$, for a more comprehensive comparison.
- We have conducted more extensive testing of our method:
- We added an experiment to test the performance of our method over a longer forecast time horizon. The experiment settings, results, and findings are included in Appendix B.2.1.
- We conducted another experiment to evaluate the time and memory usage of the MG-TSD model during training. The experiment settings, results, and findings are included in Appendix B.2.2.

We have carefully proofread our manuscript and corrected any typos, as well as imprecise labels or captions in the plots.

We are intensively working on revisions. Our code will be made available to the public soon.

---

### Meta-Review · Area_Chair_3Nzs · 2023-12-09

**Metareview:**

The paper presents an intriguing approach to time series forecasting with the MG-TSD model, leveraging diffusion models and multi-granularity data. The strengths lie in the innovative methodology, clear presentation, and robust experiments. The weaknesses, such as limited metric diversity and unclear performance improvement over baselines, need to be addressed for a more comprehensive evaluation in the final version of the draft. Furthermore, comparisons with relevant existing works and Transformer-based models would enhance the paper's contribution and practical relevance.

**Justification For Why Not Higher Score:**

1. Some related works could be further discussed, and additional metrics in the main experiment could provide a more comprehensive evaluation.
2. The performance improvement over baseline models is not as pronounced as expected, and the use of the RNN architecture requires further explanation.

**Justification For Why Not Lower Score:**

All reviewers reached a consensus to (marginally) accept this paper.

---

### Decision · Program_Chairs · 2024-01-16

Accept (poster)